# A neural circuit for wind-guided olfactory navigation

Andrew M. M. Matheson [1,4], Aaron J. Lanz [1], Ashley M. Medina[1], Al M. Licata[1], Timothy A. Currier [1,2,5], Mubarak H. Syed[3] & Katherine I. Nagel [1] ✉

To navigate towards a food source, animals frequently combine odor cues about source identity with wind direction cues about source location. Where and how these two cues are integrated to support navigation is unclear. Here we describe a pathway to the *Drosophila* fan-shaped body that encodes attractive odor and promotes upwind navigation. We show that neurons throughout this pathway encode odor, but not wind direction. Using connectomics, we identify fan-shaped body local neurons called hΔC that receive input from this odor pathway and a previously described wind pathway. We show that hΔC neurons exhibit odor-gated, wind direction-tuned activity, that sparse activation of hΔC neurons promotes navigation in a reproducible direction, and that hΔC activity is required for persistent upwind orientation during odor. Based on connectome data, we develop a computational model showing how hΔC activity can promote navigation towards a goal such as an upwind odor source. Our results suggest that odor and wind cues are processed by separate pathways and integrated within the fan-shaped body to support goal-directed navigation.

Searching for a resource such as food requires the integration of multiple sensory cues. In natural environments, food odors are often transported by wind, forming turbulent plumes[1,2]. Within these plumes, instantaneous odor concentration is often a poor cue to source direction[3–5]. Thus, many organisms have evolved a strategy of using odor information to gate upwind (or upstream) movement to locate the source of an attractive odor[6–10]. This strategy complements those observed in odor gradients, where odor increases drive straighter trajectories, while odor decreases drive re-orientation (e.g.[11]). Navigation towards potential food sources thus requires integration of directional information about the prevailing wind (often derived from mechanosensation), with information about the identity and quality of odors carried on that wind[9,10]. Where and how these two types of information are integrated to support navigation towards an odor source is not clear.

In the insect brain, several conserved central neuropils have been implicated in olfactory food search and navigation (Fig. 1A). The mushroom body (MB) and lateral horn (LH) have been implicated in learned and innate olfactory processing, respectively[12–14]. Subsets of MB and LH output neurons (MBONs and LHONs) promote approach and avoidance behavior[15–18]. A number of putative mechanosensory inputs to the MB have been identified[19] and wind intensity signals have been observed in certain MB compartments[20]. The LH also receives input from mechanosensory centers in a discrete ventral region[17,19]. However, it is not known whether the MB and LH represent wind direction signals, as well as odor identity and value signals, to support navigation.

In contrast, the fan-shaped body (FB), a part of the *Drosophila* navigation center called the central complex (CX), has been recently shown to encode wind direction[21]. Columnar inputs to the fan-shaped

[1]Neuroscience Institute, NYU Medical Center, 435 E 30th St., New York, NY 10016, USA. [2]Center for Neural Science, NYU, New York, NY, 4 Washington Place, New York, NY 10003, USA. [3]Department of Biology, 219 Yale Blvd NE, University of New Mexico, Albuquerque, NM 87131, USA. [4]Present address: Department of Biological Sciences, Columbia University, 600 Sherman Fairchild Center, New York, NY 10027, USA. [5]Present address: Department of Neurobiology, Stanford University, 299W. Campus Drive, Stanford, CA 94305, USA. ✉e-mail: katherine.nagel@nyumc.org

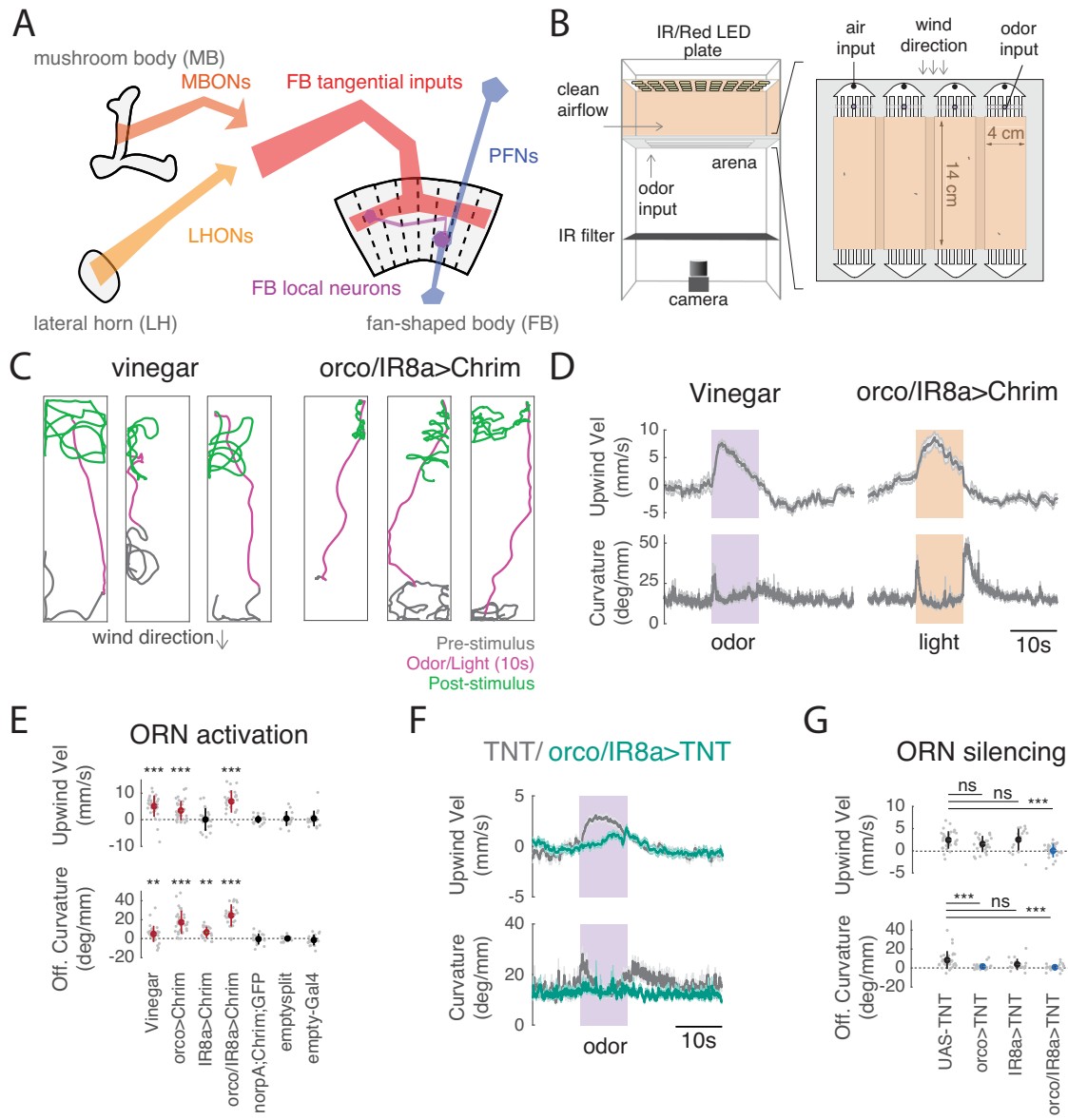

**Fig. 1 | A behavioral paradigm for investigating odor-evoked wind navigation.**
**A** Brain regions and neuronal classes investigated in this study. The mushroom body (MB) and lateral horn (LH) are higher-order olfactory centers involved in learned and innate olfactory processing, respectively. The fan-shaped body (FB) is part of the fly navigation center called the Central Complex (CX). Output neurons of the MB and LH (MBONs and LHONs) provide input directly or indirectly to FB tangential inputs. Columnar PFNs provide wind direction input to the FB. FB local neurons receive input both from FB tangential inputs and from columnar PFNs. **B** Schematic of top and side view of the behavioral apparatus showing IR illumination (850 nm), red activation light (626 nm, 26 µW/mm²), imaging camera, behavior chambers, and air and odor inputs. **C** Navigation behaviors evoked by odor and by optogenetic stimulation of olfactory receptor neurons. Example walking trajectories in response to 1% vinegar (left) and optogenetic activation of orco+ and IR8a+ ORNs (right), before (gray), during (magenta), and after (green) 10 s of odor (left) or light (right). Constant wind at -12 cm/s. **D** Time course of upwind velocity and curvature (angular/forward velocity) in response to odor or optogenetic stimulation, averaged across flies (mean ± SEM, vinegar N = 26 flies, ORN activation N = 24 flies). Shaded area: stimulation period, 10 s vinegar (purple) or light (orange). **E** Upwind velocity and OFF curvature (average change from baseline for single flies) in response to stimulation for each genotype/condition. Mean ± STD overlaid; red indicates a significant increase. Vinegar (N = 31),

orco>Chrimson (N = 31), and or/IR8a > Chrimson (N = 24) stimulation all drove significant increases in upwind velocity (0–5 s after stimulus ON, vinegar: p = 1.0997e−05, orco: p = 3.8672e−05, orco,IR8a: 2.6691e−05) and OFF curvature (0–2 s after stimulus OFF, vinegar: p = 1.9619e−04, orco: p = 1.1742e−06, orco,IR8a: p = 2.3518e−05). Light activation of flies carrying only the parental effector (nor-pA;UAS-Chrimson, N = 16), empty-GAL4 > Chrimson (N = 19), or empty-split GAL4 > Chrimson (N = 14) did not increase upwind velocity (parent: p = 0.7174, empty-split: p = 0.6874, empty-gal4: p = 0.6698) or OFF curvature (parent: p = 0.7960, empty-split: p = 0.3144,empty gal4: p=0.7354). IR8a > Chrimson stimulation did not increase upwind velocity (p = 0.3507) but did increase OFF curvature (p = 4.4934e−04). All statistics used two-sided Wilcoxon signed rank test. **F** Time course of upwind velocity and curvature in response to odor in flies with ORNs silenced (orco/IR8a > TNT, mean ± SEM, N = 26, teal) versus control (UAS-TNT, N = 31, gray). **G** Upwind velocity and OFF curvature for silencing experiments, quantified as in **E**. Mean±STD overlaid. Blue overlay represents significant decrease compared to control (UAS-TNT). (Two-sided Mann–Whitney U test compared to UAS-TNT (N = 31) control, upwind velocity: orco (N = 25): p = 0.10627, ir8a (N = 18): p = 0.40095, orco,ir8a (N = 26): p = 0.00010917, OFF curvature: orco: p = 3.2758e−05, IR8a: p = 0.037135, orco,IR8a: p = 4.2482e−05). All statistics corrected using the Bonferroni method. Source data for panels that show statistical tests for this and all subsequent figures are provided as a Source Data file.

body (FB), known as PFNs, represent wind direction as a set of orthogonal basis vectors, and receive input from the lateral accessory lobe (LAL) via LNa neurons[21]. Wind direction signals are strongest in PFNs targeting ventral layers of the FB (PFNa, p, and m in the hemibrain connectome[22]). PFNs targeting more dorsal layers (PFNd and v) encode both optic flow[23] and self-motion during walking[24] in a similar vector format. PFNs of all types show little sensitivity to odor stimuli[21] suggesting that this pathway mostly encodes flow and self-motion information independent of odor.

A distinct set of FB inputs, known as FB tangential cells, are anatomically downstream of the MB[25,26] however, most of these have not been functionally characterized. To date, few olfactory inputs to the FB have been described[27]. Large lesions to the FB disrupt visual navigation[28] and activation of subsets of FB neurons can produce oriented locomotor behaviors in cockroaches[29]. Recent theoretical and experimental work suggests that FB circuitry is optimized for encoding vectors and specifying navigational goals[23,24,30–32] but experimental evidence for goal encoding in the FB is still sparse. Numerous studies have explored the role of the FB in path integration[31,33] visual navigation[34] and landmark-guided long-distance dispersal[35–37]. However, few studies have investigated the role of this region in olfactory navigation[38].

Here we used an optogenetic wind-navigation paradigm, together with calcium imaging, connectomic analysis, and computational modeling to ask how the MB/LH and FB work together to promote olfactory navigation behavior. We show that a subset of attraction-promoting MB and LH neurons evoke upwind movement when activated. However, calcium imaging indicates that these neurons do not strongly encode wind direction, suggesting that integration of odor and wind information occurs elsewhere. We next performed a large behavioral screen of FB inputs, finding that several groups of FB tangential inputs, but not columnar PFNs, drive upwind movement. Imaging revealed that these neurons also encode odor but not wind direction. Finally, we identify a specific type of FB local neuron, called hΔC, that receives input from both wind-sensitive PFNs and from odor-sensitive FB tangential cells. We show that these neurons encode an odor-gated wind direction-tuned signal, and promote navigation in a reproducible direction when sparsely activated in a fly-specific pattern. Silencing hΔC neurons impairs the ability of flies to maintain upwind orientation throughout an odor stimulus. Based on motifs from the fly connectome, we develop a computational model showing how different patterns of activity in hΔC neurons can promote navigation either upwind (under natural odor and wind activation) or in a reproducible arbitrary direction (during sparse optogenetic activation). Taken together, our data support an emerging model of the FB, in which spatial direction cues and non-spatial context cues enter this region through distinct anatomical pathways, and are integrated by local neurons to specify goal-directed navigation behaviors.

## Results

### An optogenetic paradigm to investigate the neural circuit basis of upwind navigation

To investigate the neural circuit basis of wind-guided olfactory navigation, we developed an optogenetic activation paradigm. We modified a set of miniature wind tunnels (Fig. 1B[10]) to present temporally controlled red light stimuli as walking flies navigated in a laminar wind flow. To validate this assay, we asked whether optogenetic activation of olfactory receptor neurons (ORNs) with Chrimson could produce behavioral phenotypes similar to those observed with an attractive odor, apple cider vinegar (vinegar). We found that broad activation of ORNs using either the orco, or the orco and IR8a co-receptor promoters together[39,40] resulted in robust navigation behaviors similar to those observed with vinegar (Fig. 1C, D, Fig. S1A). In response to either odor or light, flies ran upwind, generating an increase in upwind velocity. Following odor or light OFF, flies initiated a local search,

characterized by increased curvature. Neither behavior was observed in the absence of the orco-GAL4 or orco/IR8a-GAL4 driver, or when we expressed Chrimson under an empty-GAL4 or empty split-GAL4 driver (Fig. 1E, Fig. S1B). Silencing both orco and IR8a-positive ORNs using tetanus toxin abolished both upwind and search responses to odor (Fig. 1F, G). Thus, optogenetic activation can substitute for odor in producing both upwind orientation and OFF search, and ORNs are required for these behavioral responses to odor.

Vinegar activates a subset of both orco+ and IR8a+ glomeruli[41]. Although the behavioral phenotypes evoked by vinegar and by optogenetic activation of ORNs were similar, they exhibited some subtle differences. Vinegar produced a stronger upwind response than optogenetic activation of orco+ ORNs in the same flies (Fig. S1A). However, the OFF search behavior evoked by optogenetic activation orco-GAL4, or of orco/IR8a-GAL4, was more robust than that evoked by vinegar (Fig. 1C, D, Fig. S1A). Moreover, activation of orco/IR8a+ ORNs in the absence of wind produced OFF search without upwind orientation (Fig. S1C). These results indicate that upwind orientation can be evoked independently of OFF search, and suggest that these two behaviors are driven by distinct but overlapping populations of olfactory glomeruli. Activation of single ORN types known to be activated by vinegar[41] did not generate significant upwind orientation or OFF search (Fig. S1D). In addition, silencing of orco+ or IR8a+ ORNs alone did not abolish upwind orientation, but did reduce OFF search (Fig. 1G, Fig. S1E). These data indicate that groups of ORNs must be activated together to promote upwind orientation and that substantial silencing of most olfactory neurons is required to abolish upwind movement in response to vinegar.

### A subset of LHONs and MBONs drive wind navigation and encode a non-directional odor signal

We next asked whether activation of central neurons in the LH and MB could similarly produce wind navigation phenotypes. We activated several groups of LHONs and MBONs that were previously shown to produce attraction or aversion in quadrant preference assays[15,17]. We found that several of these neuron groups drove robust upwind movement when activated (Fig. 2A, B, Fig. S2A). Neurons promoting upwind movement included the cholinergic LHON cluster AD1b2 (labeled by LH1396, LH1538, and LH1539 (Fig. 2A), and the cholinergic MBON lines MB052B (labeling MBONs 15–19), MB077B (labeling MBON12), and MB082C (labeling MBONs 13 and 14, Fig. 2B). AD1b2 drivers and MB052B also elicited significant increases in OFF curvature when activated (Fig. 2A, B), while activating individual MBONs within MB052B (MBONs 15–19) did not drive significant upwind movement (Fig. S2B). Silencing single MBON or LHON lines that drove navigation phenotypes did not abolish upwind movement in response to odor (Fig. S2C) consistent with models suggesting that odor valence is encoded by population output at the level of the MB[16] and with our findings at the periphery that very broad silencing is required to eliminate behavioral responses to vinegar.

We also identified MBONs that produced other navigational phenotypes. For example, the glutamatergic (inhibitory) MBON line MB434B (labeling MBONs 5 and 6), which was previously shown to produce aversion[15] generated downwind movement in our paradigm (Fig. 2C). Moreover, two MBON lines produced straightening (reduced curvature) in our paradigm (Fig. 2D) but no change in movement relative to wind (Fig. 2D, Fig. S2A): the GABAergic line MB112C (labeling MBON 11), which evoked attraction in quadrant assays, and the glutamatergic line MB011B (labeling MBONs 1,3,4), which evoked aversion[15]. Overall, these results indicate that LH/MB outputs can drive coordinated suites of locomotor behavior that promote attraction or aversion in different environments. Several LHONs and MBONs redundantly drive upwind movement, key to attraction in windy environments, while other MBONs drive straightening, which promotes attraction in odor gradients[11] or in response to familiar visual stimuli[42].

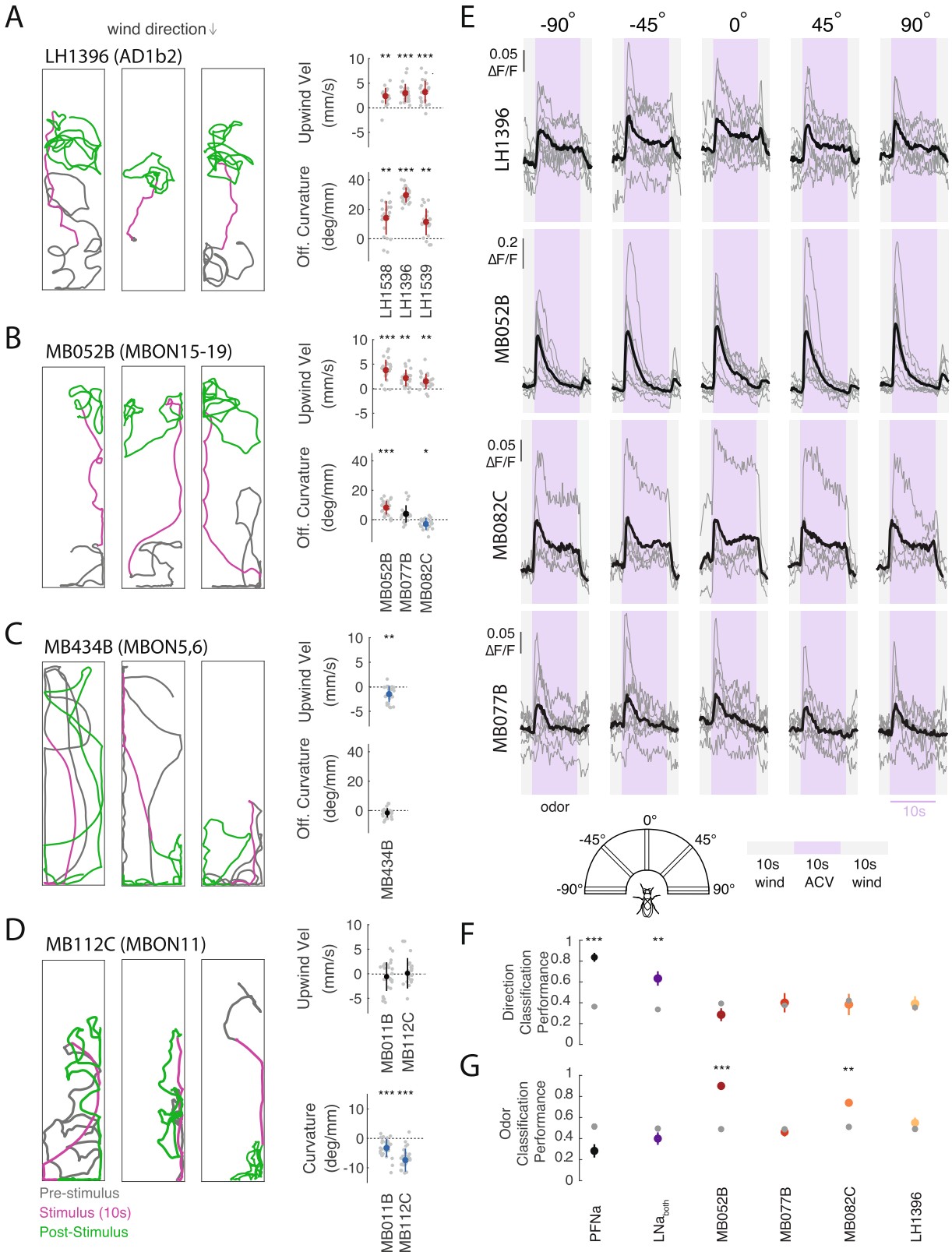

Are the LHONs and MBONs that drive upwind movement sensitive to wind direction, or do they encode a non-directional odor signal that is integrated with wind direction downstream? To answer this question, we used calcium imaging to measure responses to calibrated wind and odor stimuli delivered from five different directions (Fig. S2D). Across all four upwind-promoting lines (MB052B, LH1396, MB077B, and MB082C), we observed responses to vinegar, but no

tuning for wind direction (Fig. 2E, S2E, ANOVA: $F_{(4,35)} = 0.35$, $p = 0.8408$, $F_{(4,40)} = 0.3$, $p = 0.8794$, $F_{(4,25)} = 0.2$, $p = 0.9989$, $F_{(4,35)} = 0.67$, $p = 0.6166$). We separately used electrophysiology to show that the α'3 compartment of the MB, which was previously shown to respond to airflow[20] does not encode wind direction (Fig. S2F). To rigorously test whether MB/LH responses carry wind direction information, we generated tree classifiers (see Methods) and asked them to

**Fig. 2 | LH and MB output neurons promote wind navigation behavior but encode odor independent of wind direction. A** Optogenetic activation of AD1b2 LHONs drives upwind movement and OFF search. Left: Example behavioral trajectories driven by optogenetic activation of AD1b2 neurons labeled by LH1396 (left). Right: Upwind velocity and OFF curvature (as in Fig. 1E) for three lines labeling AD1b2 LHONs: LH1538 ($N = 21$ flies), LH1539 ($N = 22$), LH1396 ($N = 24$). All three lines significantly increase both upwind velocity and OFF curvature (upwind: $p = 2.4548e$ $-04$, $7.9941e-05$, $1.8215e-05$; OFF-curvature: $p = 3.6712e-04$, $1.7699e-04$, $1.8215e-05$ respectively). **B** Optogenetic activation of attraction-promoting MBONs drives upwind movement and OFF search. Left: Example behavioral trajectories driven by optogenetic activation of MBONs 15–19 labeled by MB052B (left). Right: Upwind velocity and OFF curvature for three cholinergic MB lines: MB052B ($N = 27$), MB077B ($N = 21$), and MB082C ($N = 24$). Each line labels distinct MBONs. All increase upwind velocity ($p = 1.0997e-05$, $1.2267e-04$, $2.0378e-04$) while MB052B increases OFF curvature ($p = 6.2811e-06$), MB077B does not ($p = 0.0046$) and MB082C reduced curvature ($p = 0.0018$). **C** Activation of aversion-promoting MBONs promotes downwind movement. Left: Example behavioral trajectories in response to optogenetic activation of glutamatergic MBONs 5 and 6, labeled by the line MB434B (left). Right: Upwind velocity and OFF curvature for MB434B ($N = 24$). MB434B significantly decreases upwind velocity ($p = 2.2806e-04$) but not OFF curvature ($p = 0.0258$). **D** MBONs promoting straighter trajectories. Example behavioral trajectories in response to optogenetic activation of the GABAergic

MBON11, labeled by the line MB112C (left). Right: Curvature during stimulus (from 2 to 5 s after stimulus ON) for MB112C ($N = 29$), and MB011B ($N = 28$). MB112C and MB011B significantly reduce curvature during the stimulus (MB112C: $p = 3.5150e$ $-06$, MB011B: $p = 3.407e-05$). **E** Upwind-promoting LHONs and MBONs encode odor independent of wind direction. Calcium responses ($\Delta F/F$) measured in four lines that all drove upwind movement. Responses were measured in LH dendritic processes of LH1396 ($N = 8$ flies), in output processes of MB052B ($N = 9$ flies), MB082C ($N = 5$), MB077B ($N = 8$). All responses measured using GCaMP6f in response to odor (10% vinegar, purple) and wind (gray) delivered from five directions (schematic). Gray traces represent individual flies, black traces represent mean across flies. **F** Performance of a wind direction (left, center, right) tree classifier trained on the first 5 s of the odor period. Gray dots represent a classifier trained with the same data and shuffled labels. PFNa ($N = 11$) $p = 3.5063e-09$, LNa ($N = 5$) $p = 5.3035e-04$, MB052B ($N = 9$) $p = 0.1044$, MB077B ($N = 8$) $p = 0.7911$, MB082C($N = 5$) $p = 0.7116$, LH1396 ($N = 8$) $p = 0.6229$. **G** Performance of an odor versus wind classifier trained on the first 5 s of wind or odor. Gray dots represent a classifier trained with the same data and shuffled labels. PFNa ($N = 11$) $p = 0.0657$, LNa ($N = 5$) $p = 0.3231$, MB052B ($N = 9$) $p = 2.7204e-08$, MB077B ($N = 8$) $p = 0.4104$, MB082C ($N = 5$) $p = 4.2903e-04$, LH1396 ($N = 8$) $p = 0.0909$. All statistics in **A**–**D** used two-sided Wilcoxon signed rank test and show mean ± STD. Classifiers in **F, G** used two-sided Student's $t$-tests and show mean ± SEM. All statistics corrected using the Bonferroni method.

decode if wind was presented from the left, right, or center relative to the fly (Fig. 2F) based on responses during the odor period. We found that classifiers trained on MB/LH responses performed no better than shuffled controls (Fig. 2F). In contrast, classifiers trained on the responses of wind pathway neurons (PFNa, or the difference between left and right LNa neurons)[21] performed significantly better than shuffled controls. We trained a second set of classifiers to discriminate between odor and wind ON (Fig. 2G). In this case, MB052B and MB082C performed significantly better than control, while neither wind pathway neuron showed significant discrimination. All neurons significantly discriminated odor from baseline (Fig. S2G). We also applied our wind direction classifiers to other phases of the response, such as wind ON and OFF (Fig. S2H). LH1396, but no other neuron group or phase, showed some discrimination at wind ON. This was largely due to responses that were stronger in front of the fly than at the sides. Taken together, these analyses support the idea that MB/LH neurons that promote upwind orientation largely encode odor presence independent of wind direction.

## Multiple tangential FB inputs promote upwind orientation and respond to odor

The FB is anatomically downstream of the MB and LH[25,26] and has previously been shown to encode wind direction. We, therefore, asked whether inputs to the FB are likewise capable of driving movement relative to wind direction. We first confirmed that the FB is anatomically downstream of our neurons of interest by performing anterograde trans-synaptic tracing[43] on two of our lines that drove upwind movement (MB052B and LH1396); we observed signal in the dorsal layers of the FB in both cases (Fig. 3A).

To ask whether the FB plays a role in wind-guided navigation, we performed an activation screen of ~40 lines labeling FB input neurons, including dorsal and ventral FB tangential inputs, and columnar PFNs, as well as additional CX neurons (Fig. 3B, Fig. S3). We performed this screen using genetically blind flies (see Methods) and in the presence of teashirt-Gal80[44] to reduce potential Chrimson expression in the ventral-nerve cord (VNC) (see Methods). We found that 4 lines labeling FB tangential inputs, but no lines labeling PFNs, generated significant movement upwind (Fig. 3B, S4A). Two dorsal FB input lines that were previously shown to promote sleep (23E10 and 84C10[45]) did not produce any wind-oriented movement in our assay, although we did observe a decrease in groundspeed in 23E10. FB tangential lines driving upwind phenotypes targeted both dorsal and ventral layers of the

FB. We attempted to refine these lines by making split-Gal4 drivers from combinations of these hemidrivers, but only one of these, labeling a set of ventral FB tangential inputs, also drove an upwind phenotype (Fig. 3C). Most split-Gal4 drivers labeled only a very small number of neurons.

In addition to the lines identified through our screen, we identified the neuron FB5AB using the connectome (Fig. 3D, E). This single pair of neurons stood out as the only FB input that receives at least one direct synaptic input from each of the MBON lines with wind navigation phenotypes (Fig. 3D). In addition, FB5AB receives the largest number of di-synaptic LH inputs from vinegar-responsive glomeruli of any FB input neuron (Fig. 3E, see Methods). We identified a GAL4 line that labels FB5AB neurons (21D07). As 21D07 labels some neurons in the antennal lobe, a primary olfactory area, we used the cell class-lineage intersection (CLIN[46]) technique to limit the expression of Chrimson to neurons in the FB (see Methods). High intensity light activation of this driver, which weakly but specifically labels FB5AB, also drove upwind orientation (Figs. 3C, 4A, B).

Overall, upwind velocity responses to FB tangential input stimulation were more persistent than those evoked by optogenetic activation in the MB and LH, continuing to promote upwind displacement after light OFF, rather than evoking search behavior (Figs. 3F, 4A, S4A). We characterized the neurotransmitter phenotypes of each of the hits from our screen and found that all were cholinergic, and thus excitatory (Fig, S4B). As in the MB and LH, silencing of individual FB input lines that drove upwind movement was not sufficient to block upwind movement in response to odor (Fig. S4C). Together these results support the hypothesis that patterns of population activity in FB tangential inputs can promote upwind movement.

We next sought to characterize the sensory responses of upwind-promoting FB tangential inputs (Fig. 4A, B) by performing calcium imaging in response to wind and odor from different directions (Fig. 4C, S4D). We observed responses to vinegar in all but one line (45D04). No FB tangential line showed significant directional tuning (ANOVA: 21D07: $F_{(4,40)} = 2.14$, $p = 0.0938$, 65C03: $F_{(4,30)} = 0.68$, $p = 0.6096$, 12D12: $F_{(4,25)} = 0.13$, $p = 0.971$, vFB split: $F_{(4,25)} = 0.64$, $p = 0.6358$) nor were tree classifiers trained on their odor responses able to distinguish between odor delivered from the left, right or center (Fig. 4D, S4E). In contrast, tree classifiers trained to distinguish odor ON from wind ON performed better than shuffled controls for 2 of 4 lines (vFB split and FB5AB, Fig. 4D), while all performed better than shuffled control when trained to distinguish odor from baseline

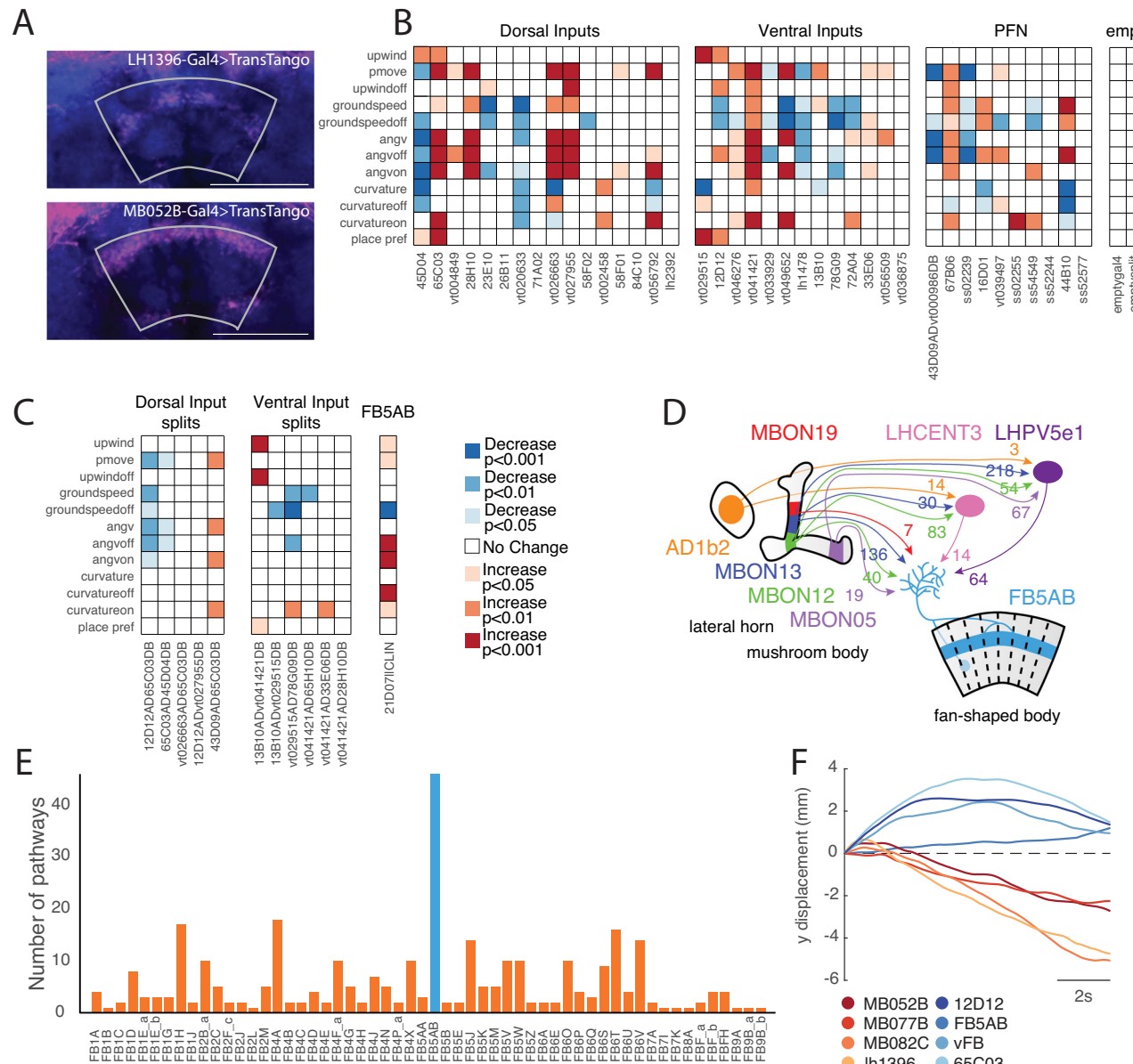

**Fig. 3 | A set of FB tangential inputs promote wind navigation behavior.**
**A** Trans-synaptic tracing reveals connections between upwind-promoting MB/LH neurons and FB tangential neurons. *Trans-tango* signal driven by LH1396-GAL4 (top) and MB052B-GAL4 (bottom). Trans-synaptic signal (magenta) was observed in horizontal layers of the dorsal FB in both cases. Neuropil is shown in blue. The FB is outlined in gray. Scale bar 50 μm. **B** Optogenetic activation results for FB inputs, including dorsal tangential inputs, ventral tangential inputs, columnar PFNs, and empty-GAL4 and empty-split-GAL4 controls. Two dorsal inputs and two ventral inputs drove significant increases in upwind velocity. Control lines drove no significant change in any measured behavioral parameter (see Methods). **C** Optogenetic activation results for split GAL4 lines labeling dorsal and ventral tangential FB inputs, and for a line labeling FB5AB (21D07-GAL4||CLIN). One split-GAL4 line labeling ventral FB tangential inputs drove a significant increase in upwind velocity. Legend applies to **B** and **C**. **D** Schematic showing feedforward connectivity onto FB5AB from three upwind-promoting MBONs (MBON 19, MBON 12, MBON 13), one upwind-promoting LHON (AD1b2), and one downwind-promoting MBON

(MBON05). Pathways converge onto FB5AB directly or indirectly through LHCENT3 and LHPV5e1. Numbers represent the average synaptic weight between each cell type and the right-sided LHCENT3 (id: 487144598), LHPV5e1 (id: 328611004), or right-sided FB5AB (id: 5813047763). **E** Number of parallel lateral horn pathways from vinegar-responsive projection neurons (estimated from ORN responses in[41]) to each FB tangential input neuron. Pathways consist of two synapses: projection neuron to lateral horn neuron, and lateral horn neuron to FB tangential input neuron. Blue bar represents the number of pathways converging onto FB5AB.
**F** Upwind displacement responses to optogenetic activation of FB tangential input lines outlast the stimulus while responses to activation of MB/LH lines do not. Timecourses of average relative y-displacement (arena position) across flies, following stimulus OFF for each line. Individual fly's average positions across trials were set to 0 and relative change in position for 10 s following stimulus OFF was averaged across flies for each genotype. All statistics in **B**, **C** used two-sided Wilcoxon signed rank test and corrected using the Bonferroni method. Legend displays equivalent uncorrected alpha level.

(Fig. S4F). The largest odor responses were observed in FB5AB (21D07), although these (but not other FB tangential line responses) decayed over trials (Fig. 4E). In the line 65C03, we observed odor responses only from the dorsal layers of the FB, while in the line 12D12, we observed

odor responses only from the ventral layers of the FB. Together these data identify a population of olfactory FB tangential inputs, targeting multiple layers of the FB, that respond to attractive odor and promote upwind movement.

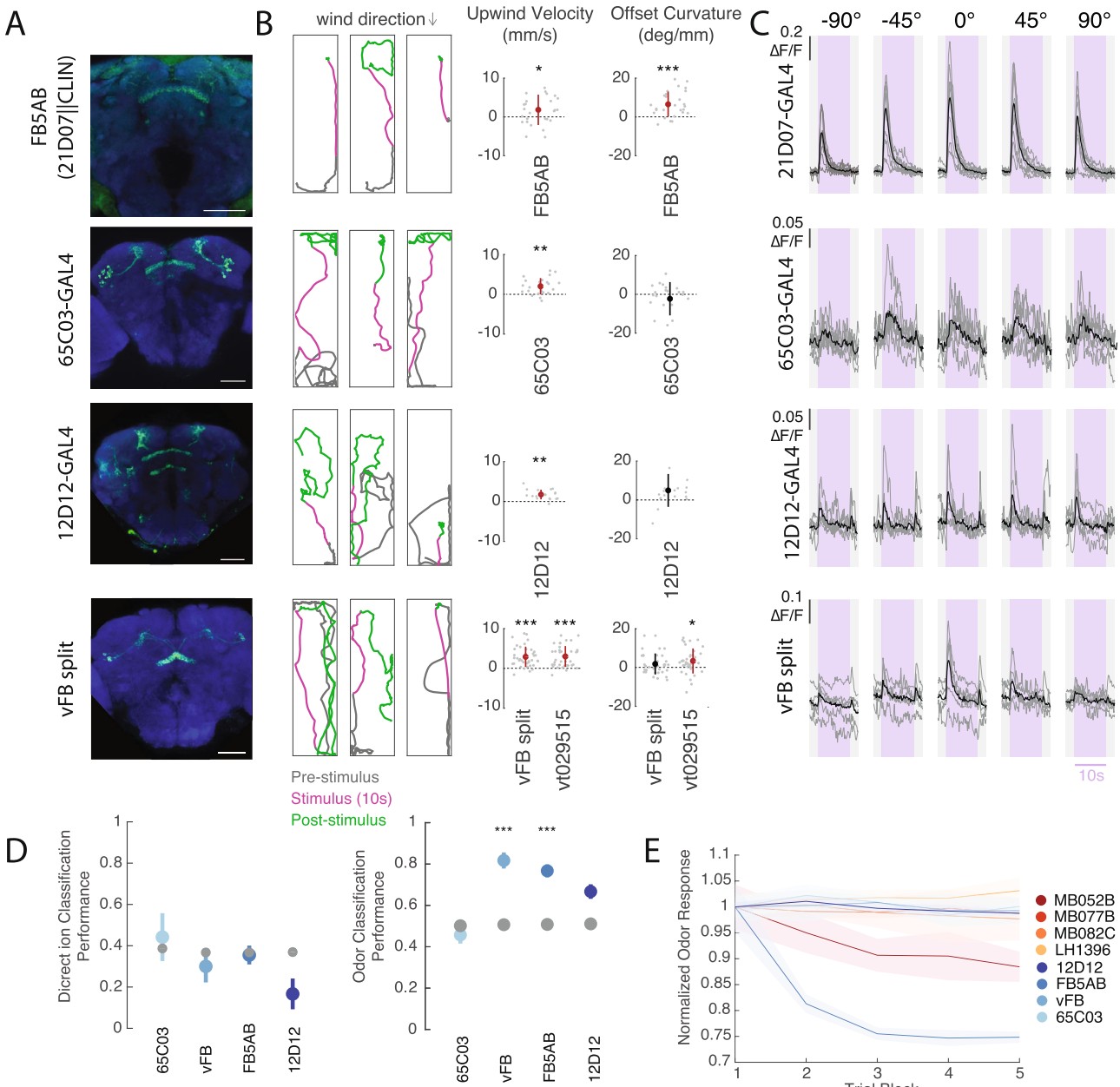

**Fig. 4 | Multiple FB tangential inputs respond to attractive odor and drive upwind movement. A** Confocal images of lines that showed FB responses to vinegar and drove upwind movement when activated. Each image shows stain for mVenus expressed with UAS-Chrimson in flies of the same genotype used for activation experiments (abbreviated genotypes shown at left). Scale bar 50 μm. **B** Example behavioral trajectories and quantification of upwind velocity and OFF curvature in each line shown at left. For FB5AB only light intensity was 34 μW/mm². For all drivers, upwind velocity was quantified over 0–10 s after odor ON. Right: upwind velocity, OFF curvature 21D07‖CLIN (*N* = 27): *p* = 0.0306, 8.1448e−05, 65C03 (*N* = 24): *p* = 4.1850e−04, 0.5841, 12D12 (*N* = 19): *p* = 1.8218e−04, 0.0055 vFB split (*N* = 38): *p* = 3.4153e−07, 0.1155, VT029515-GAL4 (*N* = 38): *p* = 4.3255e−07, 0.0024). **C** Calcium responses in each line shown at left in response to odor delivered from five different directions (as in Fig. 2E). Shaded purple region indicates odor period. Gray lines represent individual flies and black represents the mean across flies: 21D07 (*N* = 9), 65C03 (*N* = 7), 12D12 (*N* = 6), vFB split (*N* = 6 flies).

**D** Performance of tree classifiers for wind direction (left) and odor presence (right) for FB tangential inputs. Gray dots represent classifiers trained with the same data and shuffled labels. Left: Performance of wind direction (left, center, right) classifier trained on the first 5 s of odor period. 65C03 *p* = 0.6450, vFB *p* = 0.4100, FB5AB *p* = 0.7882, 12D12 *p* = 0.0177. Right: Performance of odor versus wind classifier trained on first 5 s of wind and odor ON. 65C03(*N* = 7) *p* = 0.0203, vFB (*N* = 6) *p* = 1.0934e−06, FB5AB (*N* = 9) *p* = 2.8375e−04, 12D12 (*N* = 6) *p* = 0.0383. **E** Decay of fluorescence response to odor over trial blocks. The response to each trial block was calculated as the average odor response to 5 consecutive trials (each from one of the directions), averaged across all flies of a genotype. Response magnitude was normalized by the average response to the first block for each line. Shaded area represents standard error across flies. Statistics in **B** used two-sided Wilcoxon signed rank test and show mean ± STD. Classifiers in **D** used two-sided Student's *t*-tests and show mean ± SEM. All statistics corrected using the Bonferroni method.

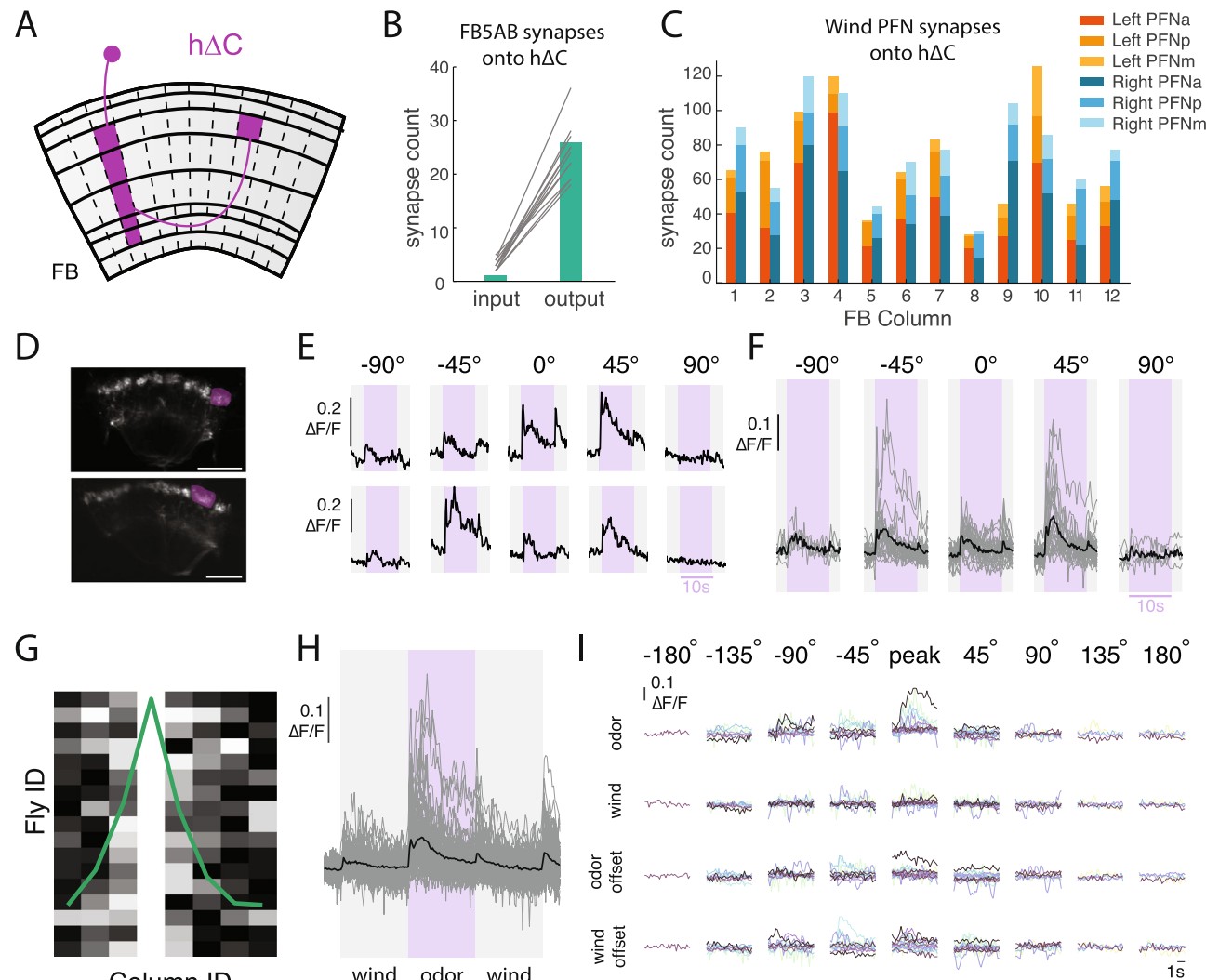

**Fig. 5 | hΔC neurons exhibit odor-modulated wind direction tuning.**
**A** Schematic of an individual hΔC neuron (purple). 20 hΔC neurons tile the FB receiving input in layers 2–6 in a single column, and projecting halfway across the FB to output in layer 6. **B** Number of FB5AB synapses onto the input and output tufts of hΔC for every FB5AB-hΔC pair. **C** Number of synapses from left and right wind-sensitive PFNs (PFNa, PFNp, and PFNm, tuned to 45° left and right respectively) onto hΔC neurons, summed within columns. **D** 2-photon image of tdTOM expressed with GCaMP6f under VT062617-GAL4 which labels hΔC neurons. Purple ROI drawn around output tuft of a single column for analysis. Scale bar 25 μm. **E** Odor responses of two example flies/columns showing directionally tuned odor responses. **F** Summary of all measured odor responses >2STD above baseline across columns and flies. Gray traces represent individual columns and black traces represent mean across columns. **G** Directional responses are restricted to nearby columns: maximally responsive direction for each fly, where data are phase shifted so maximum columns align at column 4. Each fly normalized to maximum column response. **H** Summary of wind and odor responses for all measured responses >2STD above baseline across columns and directions. Responses to odor are stronger than those to wind ON, odor OFF and wind OFF ($n = 87$ responsive columns from $N = 16$ Flies). **I** Summary of wind and odor phase activity of maximally responsive column, aligned to maximally responsive direction. Data are shifted to be aligned to the maximal direction and each row represents different stimulus period. Each color represents a different fly ($N = 16$).

## hΔC neurons encode a wind direction signal that is modulated ON by odor

Our functional imaging data suggest that, like the upwind-promoting neurons in the MB/LH, FB tangential inputs are not directionally tuned for wind. In contrast, PFNa and LNa encode wind direction, but not the presence of vinegar (Fig. 2F). Together these data suggest that columnar and tangential inputs to the FB encode directional and non-directional information, respectively. We therefore hypothesized that FB local neurons, which receive input from both columnar and tangential inputs, might integrate these two types of information.

To identify FB local neurons that integrate odor and wind direction signals, we used the hemibrain connectome[22] to search for neurons that receive input both from FB5AB and from the most wind-sensitive PFNs in our previous survey (PFNa, p, and m[21]). This analysis revealed a population of 20 hΔC neurons that tile the vertical columnar

structure of the FB (Fig. 5A[31]). Each hΔC neuron receives input from both left and right-preferring wind-sensitive PFNs at its ventral dendrites, and from FB5AB at its dorsal axons, where excitatory olfactory input might gate synaptic output through axo-axonic connections (Fig. 5B, C).

To ask whether hΔC neurons respond to wind and odor signals, we identified a GAL4 line—VT062617—that appears to label hΔC neurons (although it may also label other local neuron types), and performed calcium imaging from the FB while presenting odor and wind from different directions. Stains revealed these neurons to be cholinergic (Fig. S5A). We observed calcium responses in the dorsal FB, where hΔC neurons form output tufts (Fig. 5D, S5B). In several examples, we observed odor responses that were strongest for a particular direction of wind (Fig. 5E). hΔC responses were strongest to odorized wind from +45° and −45° (Fig. 5F), although the relationship between

wind direction and peak response was not consistent across flies (Fig. 5E). hΔC responses were typically localized to a few nearby columns (Fig. 5G), and we observed no consistent relationship between column location and preferred wind direction when pooling data across flies (Fig. S5C). Thus, the wind direction representation in hΔC neurons appears to be unique to each fly, similar to what has been observed for heading representations in compass neurons[47], and distinct from the wind representation in PFNs, which is uniformly tuned to 45° ipsilateral[21].

We next examined hΔC responses as a function of stimulus phase. Across flies and columns, the strongest and most consistent responses occurred during the odor, although weaker responses also occurred at wind ON, odor OFF, and wind OFF (Fig. 5H). To examine whether wind tuning differed across these phases, we selected the maximally responsive column for each fly, and aligned responses to the peak direction of that response, then plotted responses to other phases of the stimulus relative to that wind direction. Responses in other stimulus phases tended to be in the same direction or nearby directions to the odor response (Fig. 5I). Taken together, our results suggest that hΔC neurons exhibit a bump of activity that depends on wind direction, but whose columnar position is unique for each fly. This bump is activated most strongly during odor, although it can also appear more weakly during other phases of the stimulus.

### hΔC neurons promote diverse navigation phenotypes and contribute to persistent upwind walking

Do hΔC neurons contribute to navigation behavior? To address this question, we activated hΔC neurons optogenetically using various spatial activation patterns, and examined the resulting behavioral trajectories (Fig. 6). Because our imaging data suggest that hΔC neurons show a bump of activity during odor and wind stimulation, we first asked what behavior was produced by sparsely activating hΔC neurons. We used SPARC2-I[48] to activate a random ~15% of hΔC neurons in each fly while they walked in the presence of laminar wind. Activation of sparse subsets of hΔC neurons (Fig, 6A) caused many flies to re-orient and then walk in a specific reproducible direction (Fig. 6B, C). Each fly walked in a distinct direction, suggesting that this was specified by the particular pattern of hΔC neurons activated in that fly (Fig. 6A, B). Directions were biased towards the long axis of our arena, but equally distributed up- and downwind (Fig. 6C). Directional walking was not due to the arena walls, as fly behavior near the walls was excluded from our analysis (see Methods). To assess the significance of this directional walking, we compared the strength of orientation in hΔC > SPARC flies to empty-GAL4 > SPARC flies (Fig. 6C, D, S6A). Empty-Gal4>SPARC flies showed significantly weaker orientation behavior (Fig. 6D). To determine whether the pattern of hΔC activation was related to walking direction, we dissected and stained each hΔC > SPARC fly. However, we observed no relationship between the anatomy of the activated columns, and the direction or strength of oriented walking (Fig S6A-E). We interpret these results to suggest that sparse activation in hΔC neurons can produce a reliable heading, and that this heading is encoded in fly-specific (not fixed) coordinates.

In addition, we asked whether broad activation of hΔC neurons could drive a behavioral phenotype. For these experiments, we generated a split-GAL4 line selectively labeling FB local neurons within VT062617, and expressed Chrimson throughout this population (Fig. 6E). Activation of this line caused unstable reorientation behaviors. At medium light levels (26 μW/mm²), fly walking was interrupted by frequent left and right turns, while at higher light levels (34 μW/mm²), flies displayed tight turns that were clockwise, counterclockwise, or alternating between clockwise and counterclockwise (Fig. 6F). Across flies, these behaviors were captured by an increase in curvature with no change in upwind velocity (Fig. 6G). Similar increases in curvature were obtained using a variety of drivers for hΔC

neurons (Fig. S6F). Therefore, uniform activation of hΔC neurons causes unstable reorientation behaviors.

Finally we asked whether hΔC activity is required for upwind orientation. In preliminary experiments, we found constitutive silencing of hΔC neurons to be lethal. We therefore used the light-activated chloride channel GtACR[49] to acutely silence hΔC neurons during odor presentation. We compared the effects of acutely silencing hΔC neurons to silencing of olfactory receptor neurons using orco,IR8a and FB5AB. Consistent with our constitutive silencing results (Fig. 1F, G), we found that acute silencing of olfactory receptor neurons impaired upwind orientation throughout the odor period (Fig. 6H). In contrast, silencing of FB5AB had no effect on upwind orientation, as we observed for other FB tangential inputs (Fig. 6H, Fig. S4C). In flies with hΔC neurons silenced, we observed normal upwind orientation early in the odor period, however trajectories then deviated significantly from upwind later in the odor period (Fig. 6H, I). We conclude that hΔC activity is required for persistent upwind fixation throughout the odor stimulus.

### A computational model of hΔC neurons' contribution to navigation behavior

Our data suggest that hΔC activity contributes to persistent upwind orientation during odor and that sparse activation of hΔC neurons can promote navigation in a reproducible direction. To understand how different patterns of hΔC neuron activation can evoke diverse behavioral phenotypes, we developed a computational model. The elements of our model are all based on neurons and connection motifs (direct or indirect) found in the hemibrain connectome (Fig. 7A[22,31]), although we imagine that additional neurons and connections may play a role in natural olfactory navigation behavior.

We first asked whether odor-gated wind direction information in hΔC neurons could be used to promote upwind movement. Although the nature of the wind representation in hΔC neurons is not entirely clear from our imaging experiments, our results are broadly consistent with a bump of activity that is tuned to wind-direction in a fly-specific manner, likely reflecting the fly-specific coordinates of the compass in each animal[47]. Thus, we imagine that hΔC neurons might show a bump of activity whose location reflects the allocentric wind direction (Fig. 7B), as has previously been described for traveling direction signals in other FB local neurons[23,24]. Alternatively, hΔC might show a bump of activity related to wind direction only in the frontal hemisphere, which could be computed based on the known frontally tuned responses of wind-sensitive PFNs (Fig. S7A, B). In either case, in our model, this bump becomes active when the fly encounters odor, due to gating input from FB5AB and other FB tangential neurons, although it could, in principle, also be activated at other times.

Activity in hΔC neurons is then translated into locomotor commands by a set of previously described output neurons: PFL3 and PFL2 (Fig. 7C, see Methods). PFL3 neurons project unilaterally to the right or left LAL and are thought to drive turning[31–34,50] while PFL2 neurons project bilaterally to the right and left LAL and are hypothesized to modulate forward walking speed[31]. PFL neurons compare the representation arriving from hΔC neurons (a bump of activity flipped by 180° due to hΔC neurons projecting to columns halfway across the FB) with a shifted representation of the fly's current heading arriving from compass neurons[31,33]. The heading representation in PFL3 neurons is shifted by −90° for right PFL3s and +90° for left PFL3s. This allows PFL3 neurons to determine whether a left or right turn will bring the fly in line with the heading specified by the hΔC bump. For example, wind to the right of the fly will generate a wind bump that overlaps more with the right PFL3 heading bump than the left (Fig. 7C). This system will cause the fly to turn until its heading bump aligns with the hΔC wind bump. In parallel, PFL2 neurons cause the fly to accelerate when the heading from hΔC overlaps with the compass heading representation. This heading representation is shifted by 180°, so it will overlap the

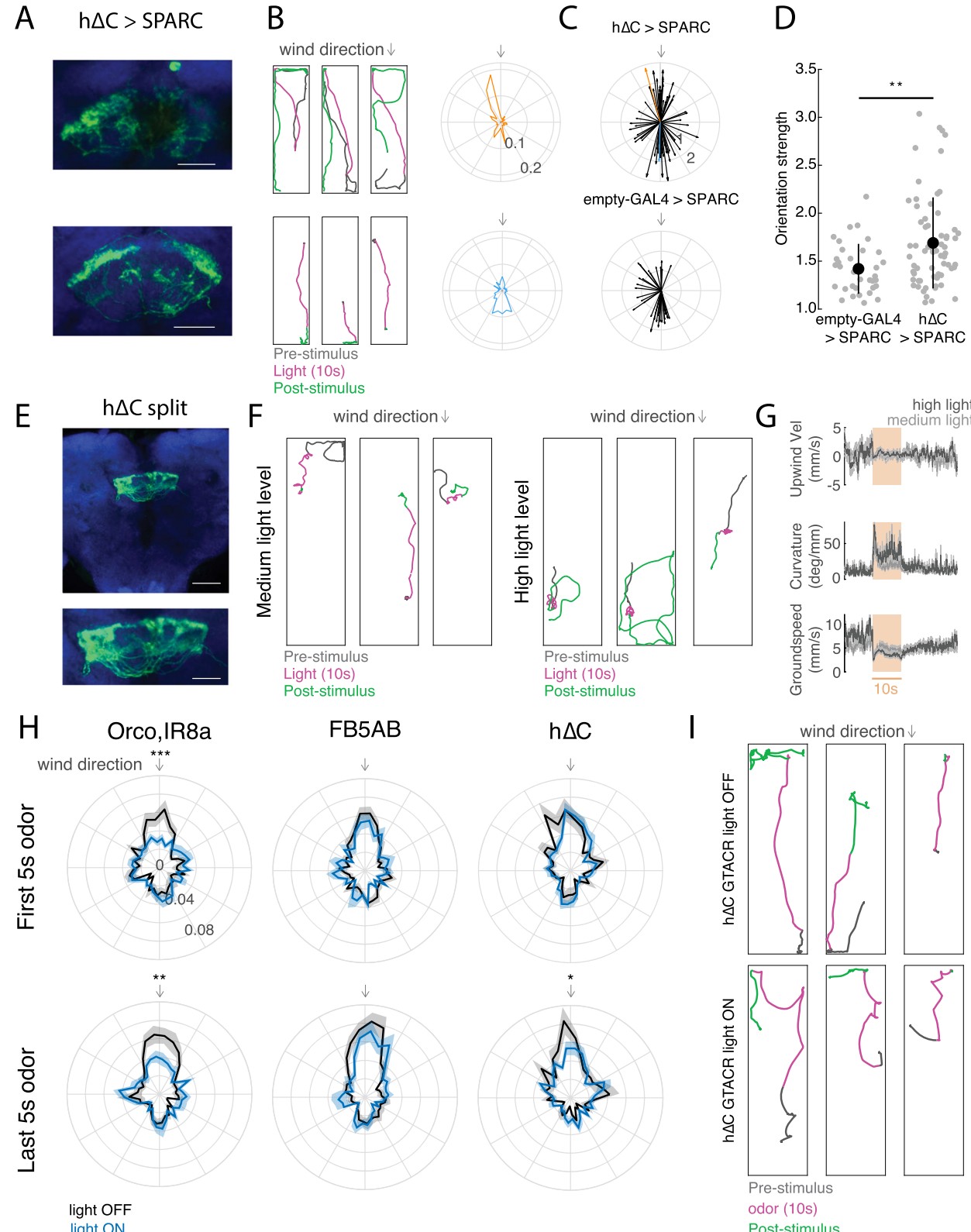

wind bump when the wind is directly in front of the fly. Thus, PFL2 neurons drive the fly to walk faster when it is pointed upwind.

We added one additional element to our model that has not been previously described. This is a second set of mutually inhibitory FB local neurons (light blue in Fig. 7C) that receive input from hΔC neurons and also send information to the PFL output system. This motif is not observed in hΔC neurons but reciprocal connections between

opposing columns can be seen in several other FB local neuron types downstream of hΔC, such as hΔA, hΔG, hΔH, and hΔM, which all send strong projections to PFL3. As we show below, this motif is required to account for the unstable reorientation behavior we observe during broad activation of hΔC neurons.

We first asked whether this simplified FB model could promote upwind movement when the hΔC wind bump is activated (Fig. 7D, E).

**Fig. 6 | Role of hΔC neurons in navigation behavior. A** Anatomical and behavior data from two representative hΔC > SPARC flies. Left: confocal images of tdTomato expressed with UAS-SPARC2-I-Chrimson. Scale bar 25 μm. **B** Behavior of two representative hΔC > SPARC flies from **A**, Left: example behavioral trajectories of flies before, during, and after optogenetic activation. Right: orientation histograms for each fly for the period 2–6 s after light ON. Radial axis represents probability. **C** Preferred orientations across all flies, where the vector direction corresponds to the preferred direction and the vector strength corresponds to the orientation index (see Methods, Fig. S6A). Top: hΔC > SPARC. Representative flies from **A**, **B**) are shown in orange and blue. Bottom: empty-GAL4 > SPARC. **D** hΔC > SPARC flies ($N = 65$) show stronger oriented walking than empty-GAL4 > SPARC2 flies ($N = 34$, see Fig. S6A for Methods, mean ± STD, two-sided ranksum test $p = 0.0035$). **E** Confocal image of hΔC split > Chrimson. Scale bar 50 μm (top), and 25 μm (bottom). **F** Example behavioral trajectories driven by hΔC split activation with 26 μW/mm² light (left) or 34 μW/mm² light (right). **G** Upwind velocity, curvature, and groundspeed (mean ± SEM) across hΔC split > Chrimson flies before, during, and after optogenetic activation using 26 μW/mm² light (light gray, $N = 32$) or 34 μW/mm² light (dark gray, $N = 14$). **H** Optogenetic inactivation of hΔC neurons using GtACR disrupts persistent upwind orientation. Each plot shows orientation histograms during light-evoked silencing (blue) compared to no-light control (black) for the first 5 s of odor (top) and last 5 s of odor (bottom). Shaded regions represent SEM. Optogenetic silencing of ORNs (orco,IR8a, $N = 24$) significantly reduces the probability of orienting upwind (±10°) during both phases ($p = 9.2724e−04$ early, 0.0090 late), while silencing of FB5AB ($N = 32$) does not ($p = 0.3848$ early, $p = 0.3259$ late). Silencing of hΔC neurons ($N = 24$) reduces upwind orientation only during the late phase ($p = 0.8512$ early, $p = 0.0475$ late). Two-sided $t$-test without correction for multiple comparisons. **I** Example behavioral trajectories in hΔC > GtACR flies in response to odor. Top: blue light off (non-silenced). Bottom: blue light on (hΔC neurons silenced).

We simulated the circuit described above using rate-based neurons (see Methods), and simulated odor input by turning on the bump of wind activity in hΔC neurons. As predicted, the odor-gated wind bump in hΔC neurons generated stable upwind orientation that was robust to allocentric wind direction and initial heading (Fig. 7E). Flies also oriented upwind using a frontal wind representation in hΔC, although this was less reliable than using an allocentric representation (Fig. S7A-D). We did not observe local search after odor OFF in our simulations, consistent with our finding that FB tangential inputs do not drive this behavior.

We next asked whether the same model could recapitulate the directional walking we observed with sparse random hΔC activation. To simulate the sparse activation experiment, we replaced the hΔC wind bump with random activation of 15% of the hΔC population in a fly-specific pattern (Fig. 7F). Similar to our behavioral results, simulated flies reoriented and then walked in an arbitrary but reproducible direction, where the direction was stable for each fly. This result occurred because the random optogenetic input provided stable non-uniform input to the PFL3 neurons, and the maximum of this input (i.e., the FB columns opposite to the highest density of 'active' hΔC neurons) establishes a stable direction. These directions were distributed in 360° space (Fig. 7G) because our simulation lacked arena constraints. Oriented walking was significantly reduced when we simulated empty-GAL4 > SPARC controls by omitting hΔC activity from our simulation (Fig. 7G).

Finally we asked whether our model could reproduce the unstable reorientation we observed with broad hΔC activation (Fig. 7H). To simulate these experiments, we uniformly activated hΔC neurons at two different intensities, corresponding to the two light levels in our experiment. Like our behavioral results, low activation generated walking interrupted with left and right turns, while high activation generated tight turns that were clockwise, counterclockwise, or alternating between clockwise and counterclockwise. Consistent with our experiments, simulated flies displayed elevated curvature that depended on the light level (Fig. 7I). The unstable reorientation behaviors in our simulations stemmed from oscillations within the mutual inhibition layer of the circuit; in the absence of this layer, uniform activation did not alter turning statistics (Fig. S7E–G). Taken together, our experiments and modeling suggest that sensory representations in FB local neurons such as hΔC can provide a goal heading that promotes stable navigation towards an environmental target such as an upwind odor source.

## Discussion
### Distinct direction and context pathways for olfactory navigation
Wind-guided olfactory navigation is an ancient and conserved behavior used by many organisms to locate odor sources in turbulent environments. Despite large differences in the types of odors they seek, and in the physics of odor dispersal, very similar behaviors have been observed in pheromone-tracking moths[51] food-seeking crustaceans[8] and ants seeking both food and their nest[52]. Thus, this behavior is required in many animals for survival and reproduction, and likely mediated by conserved neural circuits.

In a previous study, we identified inputs to the FB that encode wind direction[21]. Neurons throughout this pathway, including both LNa neurons and PFNs, show activity that is strongly modulated by wind direction, but weakly modulated by odor. Other LN and PFN neurons have been shown to strongly encode optic flow direction[23,33] another cue that signals self-motion and wind direction—particularly in flight. Taken together, current data thus suggest that the columnar pathway to the FB encodes information about self-movement and environmental flow useful for recognizing where the fly is relative to its environment.

In contrast, the present study identifies an olfactory pathway to the FB that encodes odor information largely independent of the wind direction from which it is delivered. We found that neurons in both the MB (a center for associative learning) and the LH (a center for innate olfactory and multi-modal processing) are capable of driving upwind movement. However, very few of these upwind-promoting neurons exhibited wind-direction tuning, particularly during the odor period. Thus, the outputs of these regions likely represent odor identity or value signals, as has been suggested by several recent models[42,53]. We further characterized a large population of olfactory tangential inputs to the FB that likewise promote upwind movement and encode odor independent of wind direction. At least one of these neurons is anatomically downstream of upwind-promoting MB and LH neurons. These data are consistent with previous studies showing that other FB tangential inputs encode non-spatial variables such as sleep state[45] and food choice[54]. Together, these data suggest that the pathway running from the MB/LH to the FB encodes non-spatial contextual information.

The segregation of direction and context inputs to the FB that we observe here shows some similarities to the organization of visual circuits in primates and of navigation circuits in rodents. In primate vision, processing of object location and object identity are famously thought to occur in distinct pathways[55]. In the hippocampus, inputs from the medial entorhinal cortex encode spatial cues, while anatomically distinct inputs from the lateral entorhinal cortex encode non-spatial context cues, including odors[56,57]. Thus, a segregation between the computation of spatial/directional information and context/identity information may be a general feature of central neural processing in both vertebrate and invertebrate brains.

What might be the advantage of this type of organization (Fig. 8)? One hypothesis is that it allows learning about non-spatial cues to be generalized to different spatial contexts. For example, during food-guided search, innate information about which odors signal palatable food must be integrated with learned odor associations[13,14] as well as with internal state variables such as hunger[54,58]. However, the fly may

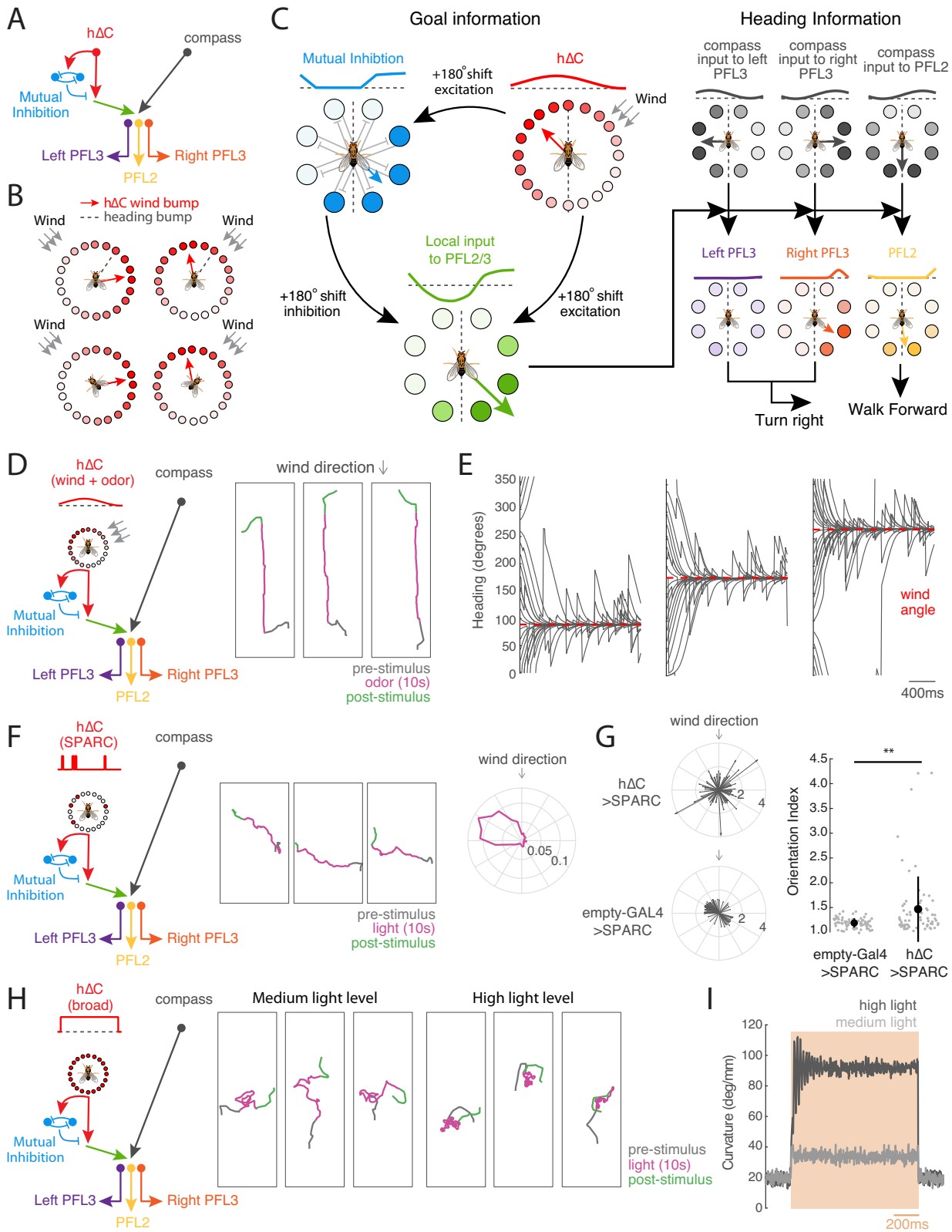

wish to generalize information learned while walking to search in flight, as has been demonstrated in bees[59]. In walking flies, wind direction can be computed directly from mechanosensors in the antennae[60,61]. In flight however, wind only transiently activates mechanosensors[62] but then displaces the fly as a whole, leading to an optic flow signal opposed to the fly's direction of movement[9,63]. As different PFNs carry

airflow and optic flow signals in a similar format[21,23] this circuit may be well-poised to compute wind direction in both walking and flying flies. Separating the computation of stimulus value, in the MB/LH and tangential FB inputs, from the computation of wind direction, in PFNs, may allow flies to generalize stimulus associations learned in one context (walking) to another (flight).

**Fig. 7 | A model of FB circuitry that translates hΔC activity into goal-directed walking. A** Overview of cell types and information flow in an FB circuit model. Odor gates a wind direction signal in hΔC neurons (red), which is processed by a mutual inhibition circuit (blue) as well as fed forward to PFL output neurons (green). PFL neurons (purple, orange, yellow) integrate goal information from hΔC with heading information from compass neurons (black) to control turning (PFL3) and forward velocity (PFL2). **B** Modeled allocentric wind representation in hΔC neurons, shown as a bump of activity (circles) and as a vector (red arrow). Heading vector in gray. Leftward rotations of the fly cause rightward rotations of the heading vector. Wind and heading vectors align when the fly is pointed upwind (bottom row). Leftward wind leads to a wind vector right of the heading vector, while rightward wind leads to a wind vector to the left (top row). An alternate wind representation is depicted in Fig. S7A. **C** Model circuit diagram showing transformation of a wind bump in hΔC neurons into upwind movement by PFL3 and PFL2 neurons. Activity patterns across each population are represented as a bump of activity across the FB (lines and circles, dotted line = 0 activity), and as a vector. An odor-gated wind bump in hΔC neurons (red) is fed forward directly to PFL2/3 neurons (green) and indirectly via a mutual inhibition circuit (blue). 180° shifts in the output of each populations transform the wind bump in hΔC into a stable bump in PFL2/3 (green). PFL3 (purple/orange) receive a heading bump from the compass system (black) shifted by 90° ipsilateral, as well as goal input from hΔC (green). Overlapping bumps lead to constructive summation, shown here for right PFL3 (orange) driving right turns. Non-overlapping bumps lead to destructive summation, shown here for left PFL3 driving left turns. Total turning represents the sum of left and right PFL3 activity.

PFL2 (yellow) receive a heading bump from the compass system (black) shifted by 180°, together with goal input from hΔC (green). Overlapping bumps lead to constructive summation that promotes faster walking. Model causes the fly to turn until the goal and heading bumps align, and to increase speed when bumps are aligned. **D** Simulated circuit activity and trajectories when odor gates the expression of an allocentric wind bump as in **B** during odor. Example trajectories (right) shown for three model flies. **E** Headings of simulated flies with different initial headings in response to odorized wind arriving from 90° (left), 180° (center), and 270° (right). Heading converges to upwind despite turning noise which drives the fly off course. **F** Simulated circuit activity and trajectories during sparse optogenetic activation (random 15% of hΔC neurons on during light). Example trajectories (center) are shown for one model fly on three different trials in which the same hΔC neurons were activated. Heading converges to the same reproducible walking direction. Orientation histogram (right) for model flies with different hΔC activation patterns. **G** Preferred orientations across simulated hΔC > SPARC flies (top) and empty-GAL4 > SPARC flies (bottom). Empty-GAL4 > SPARC (N = 72) flies were simulated by setting hΔC activity to zero. Simulated hΔC > SPARC (N = 72) flies have stronger orientation indices than simulated empty-GAL4 > SPARC flies (mean ± STD, two-sided ranksum test p = 0.0032). **H** Simulated circuit activity and trajectories during broad optogenetic activation. All hΔC neurons activated equally during light at medium (center) or high level (right). **I** Average curvature across simulated flies before, during, and after broad optogenetic activation using medium (light gray) or high light (dark gray), showing an intensity-dependent increase.

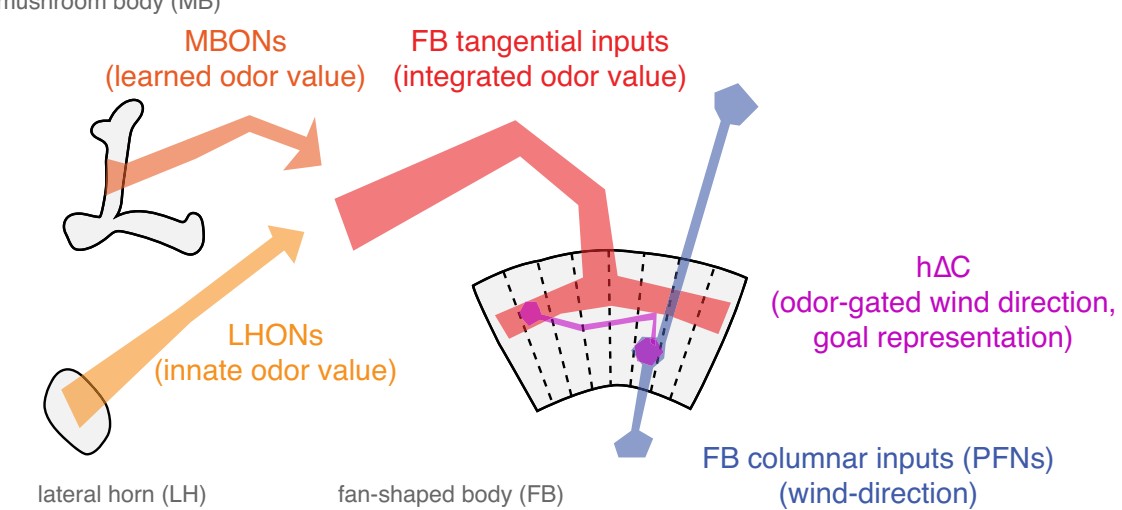

mushroom body (MB)

MBONs
(learned odor value)

FB tangential inputs
(integrated odor value)

LHONs
(innate odor value)

hΔC
(odor-gated wind direction, goal representation)

FB columnar inputs (PFNs)
(wind-direction)

lateral horn (LH)    fan-shaped body (FB)

**Fig. 8 | Conceptual model of sensory integration for olfactory navigation in the *Drosophila* central brain.** Conceptual model of central olfactory navigation circuitry as suggested by this and previous studies. MBONs, LHONs, and FB tangential inputs promote wind navigation and encode odor information but not wind direction information. FB tangential inputs are a likely locus where learned and innate odor information may be integrated to drive behavior. In contrast, FB columnar inputs (PFNs) encode wind direction but not odor presence. hΔC neurons receive input both from directionally tuned PFNs at their dendrites and from odor-tuned FB tangential inputs at their axons. They encode a fly-specific wind direction signal that can be modulated ON by odor. Sparse activation of hΔC neurons can drive movement in a reproducible direction and activity in these neurons is required for sustained upwind orientation during odor. Our data support a model in which columnar and tangential inputs to the FB encode directional and non-directional information respectively, and where tangential input can gate the expression of directional information in local neurons outputs to specify navigational goals.

## Specification of a goal direction by FB local neurons

How does the nervous system represent goals for navigation? When sensory cues are continuously available, no explicit goal representation may be required, and movement towards a goal such as food may be accomplished through chains of sensory-motor reflexes[9–11]. In contrast, when sensory cues are unavailable, such as during navigation towards a remembered shelter, the nervous system must build and store a representation of the goal location[33]. Navigation towards an odor source presents an interesting intermediate case. Food sources generally produce strong sensory cues that can be used to drive chained reflexes. However, the wind direction and odor cues produced by turbulent plumes are fluctuating, variable, and uncertain, meaning

that memory and internal estimates of source position can make an important contribution to effective navigation[64–66]. Understanding how nervous systems represent and use internal variables during this task may help us to design robust search algorithms for noisy environments.

Here we present evidence that hΔC neurons participate in specifying a goal heading during olfactory navigation. Silencing of hΔC neurons impairs persistent upwind orientation during odor, suggesting that the activity of these neurons is required to maintain upwind fixation. hΔC neurons receive input from an odor-tuned, upwind-promoting pathway through the MB/LH to FB tangential neurons. Our imaging results are consistent with a model in which hΔC neurons

represent wind direction (likely through inputs from wind-sensitive PFNs) and modulate the strength of this representation during odor (perhaps through axo-axonic gating input from FB tangential neurons). Thus, hΔC neurons are well-poised to construct an internal representation of the goal direction during olfactory navigation. However, understanding the precise nature of wind, odor, and goal representations in hΔC neurons will require future imaging of population activity during ongoing navigation.

We also show here that sparse random activation of hΔC neurons can evoke locomotion in a reproducible direction, consistent with the idea that hΔC activity patterns can specify a goal heading. We were unable to determine from our data whether hΔC activation specifies a distance (and thus a vector) as well as a direction, although many trajectories from the same fly were of similar length. Curiously, we found that hΔC activity had to be non-uniform in order to evoke directional walking, but need not be organized as a bump of activity—any asymmetry in the response across columns was sufficient. Although we produced this activity through sparse optogenetic stimulation, similar asymmetries could be generated and stored as patterns of asymmetric synaptic input from FB tangential neurons onto FB local neurons, or as asymmetric synaptic weights between interconnected FB local neurons. Because the *Drosophila* FB contains at least 27 local neuron types targeted by ~145 tangential neuron types[31] this structure could provide a reasonably large capacity for storing diverse direction or vector memories.

To understand how hΔC activity might specify a goal heading, either upwind or in an arbitrary reproducible direction, we developed a computational model. This model is based on motifs found in the *Drosophila* connectome, such as the phase shifts between compass neurons and PFL neurons[31,33] and mutual connections found in certain local neuron populations. However, the connection schemes used in our model are simplified compared to the real circuit. In our model, the architecture of the FB is used to compare a goal heading in hΔC neurons with the current heading represented in compass neurons, and then drive turning to minimize discrepancies between these directions. This computation is implemented by the convergence of hΔC input and compass neuron input onto PFL output neurons that control steering and forward velocity. Our steering model is similar to several recent models proposed for path integration and visual landmark-guided navigation[31–33]. However, our model differs in the proposed location and nature of the goal representation. In previous models, goals were stored in PFN activity[33] or in plastic synapses between compass neurons and PFLs[32]. Here the goal is instead stored in the dynamic activity pattern of a population of local neurons. One advantage of our model is that different local neurons can be rapidly switched on and off through tangential input, allowing the fly to rapidly update its goals depending on behavioral demands. A further advantage is the large number of local neurons and tangential inputs, which as noted above provide a substrate for learning, storing, and releasing multiple goal memories.

Although our results implicate hΔC neurons in the integration of odor and wind cues and the specification of a target walking direction, flies with hΔC neurons silenced were still able to initially turn upwind in response to odor, suggesting that other neurons play a role in this initial upwind turn. These might be additional FB local neuron types—we identified multiple FB tangential inputs that drive upwind movement and not all of these target hΔC. They may also include neurons that bypass the central complex. In addition to targeting the FB, MBONs also make direct connections to the lateral accessory lobe (LAL) a region implicated in motor control and steering, including odor-evoked steering behavior in moths[25,26,50,67]. As the LAL also receives wind direction input[68] it could form a second site of wind and odor integration. In this study, we found that activation of FB neurons produced more persistent wind orientation phenotypes than activation of MB/ LH

output neurons. Thus, parallel pathways from olfactory centers to motor centers could regulate behavior on different timescales. FB representations of odor and wind might also allow a fly to adopt courses at particular angles to the wind, or to generate an internal estimate of the direction of the odor source. Determining how activity in these different pathways shapes behavior on different timescales, and in different spatial contexts, will provide additional insight into the organization of central circuits for navigation.

## Methods
Our research complies with all relevant ethical regulations. Because only invertebrate organisms were used in this study, no protocol approval was required.

### Fly stocks
A complete table of fly genotypes and ages for each figure panel is given in Table S1. Information on parental stocks is given in Table S2. All experimental flies (except trans-tango flies) were raised at 25 °C on a standard cornmeal-agar medium, with a 12 h light-dark cycle. For optogenetic activation, experiments were run in genetically blind male norpA hemizygotes. All other flies were female. We previously showed no difference in olfactory behavior of male versus female flies in our assay[10]. For calcium imaging, we used older flies (5–21 days) to maximize indicator expression. For electrophysiology, we used younger flies (2–3 days) to minimize glial ensheathing. Trans-tango flies were raised at 18 °C and aged until they were 10–20 days old following recommended protocols[43].

### Behavioral experiments
Behavioral experiments were performed in a miniature wind tunnel setup modified from ref. 10 (plans and code at https://github.com/nagellab/AlvarezSalvado_ElementaryTransformations). Flies walked in a shallow acrylic arena with constant laminar airflow at ~12 cm/s and were tracked using IR LEDS (850 nm, Environmental Lights) and a camera (Basler acA1920-155um). For odor experiments, a 10 s pulse of 1% apple cider vinegar (Pastorelli) was introduced through solenoid valves (LHDA1233115HA, Lee Company) immediately below the arena. Three 70 s trial conditions (wind plus odor, wind alone, and no wind), were randomly interleaved with odor starting at 30 s. For Chrimson experiments, we used red LEDs (626 nm, SuperBrightLEDS) at an intensity of 26 μW/mm² (626 nm) for most experiments, and 34 μW/mm² for high light level and CLIN experiments, all with ambient lighting, except for SPARC experiment which were run in the dark. Four trial conditions (wind plus light pulse, light pulse alone, wind alone, blank) were randomly interleaved. For GtACR experiments, we used blue LEDs (470 nm, Environmental Lights) at an intensity of 58 μW/mm² (530 nm), and ran experiments in the dark. Four trial conditions (wind plus light pulse, wind plus odor pulse, wind plus odor and light, wind only) were randomly interleaved.

Two to seven days before the experiment, flies were collected and housed in time-shifted boxes. All flies were run between subjective ZT1-ZT4 and starved for ~24 h before the experiment. Fly food for optogenetic experiments was supplemented with 50 μL all-trans retinal (35 mM stock: Sigma, R2500, dissolved in ethanol, stored at −20 °C) mixed into ~1 teaspoon of hydrated potato flakes. Flies were briefly anaesthetized with ice for ~1 min before loading and allowed 5–10 min to recover before the experiment began.

We tracked X,Y coordinates and orientation in real time at 50 Hz using custom Labview code then analyzed data offline using Matlab (Labview code at https://github.com/nagellab/AlvarezSalvado_ElementaryTransformations:, Matlab code at https://github.com/nagellab/Mathesonetal2022). We low-pass filtered raw data at 2.5 Hz using a 2-pole Butterworth filter and removed trials where the fly moved <25 mm, and flies that moved on <5 trials for a condition. Odor

trials were aligned to the time of actual odor encounter based on PID measurements. To compute behavioral parameters as a function of time, we omitted time periods when the fly moved at <1 mm/s. Groundspeed was calculated as the distance between adjacent X,Y samples divided by the frame interval (20 ms). Upwind velocity was the difference in Y-coordinates divided by the frame interval. Angular velocity was the absolute value of the difference in unwrapped orientation divided by the frame interval. Curvature was the angular velocity divided by the groundspeed. Probability of movement (pmove) was the probability of groundspeed being above 1 mm/s. For quantifications, we compared average parameter values for each fly to a baseline period (10–25 s after trial onset). For place preference baseline was 25–30 s (immediately before stimulus ON).

We used the following time windows for analysis: pmove: 0–5 s from stimulus ON, upwind velocity (upwind): 0–5 s from stimulus ON for periphery, LH, MB, 0–10 s from stimulus ON for FB, OFF upwind velocity (upwindoff): 0–2 s after stimulus OFF, groundspeed: 2s–5s from stimulus ON, OFF groundspeed: 0–2 s after stimulus OFF, angular velocity (angv): 2s–5s from stimulus ON, OFF angular velocity (angvoff): 0–2 s after stimulus OFF, ON angular velocity (angvon): 0–1 s from stimulus ON, curvature: 2s–5s from stimulus ON, OFF curvature (curvatureoff): 0–2 s after stimulus OFF, ON curvature (curvature on): 0–1 s from stimulus ON, place preference (placepref): 7.5 s from stimulus ON to 2.5 s after stimulus OFF. For display purposes we do not depict significance values in tables (Fig. 3B, C, Fig. S3) for upwind velocity for small magnitude increases (<1 mm/s), or for curvature (ON and OFF) if angular velocity is <50 deg/s and groundspeed is <4 mm/s. In these cases, curvature increase was due mostly to a drop in groundspeed and trajectories did not exhibit characteristics of search. To measure upwind displacement (Fig. 3F) we computed displacement relative to position at stimulus OFF for each fly and show the mean across flies. For orientation histograms (Figs. 6H, 7F, S6B) we plotted the histogram of orientations that each fly adopted while moving >1 mm/s. We compared the probability of adopting an orientation ±10° from upwind between odor and odor+light conditions using a student's *t*-test.

For SPARC experiments, we analyzed the orientation index and preferred direction using data from 2–6 s from stimulus ON. We excluded all data points where the fly was within 3 mm of the arena walls as well as points where they fly moved at <1 mm/s. The orientation index was computed by taking the SVD of the orientation histogram, projecting the data onto the two largest principal components (PCs), and then calculating the ratio of the standard deviations along these two dimensions. Histograms that show more pronounced orientation exhibit higher ratios (see Fig. S6A). Preferred direction was the direction along the first PC with the highest value. Flies with fewer than 15 trials or fewer than 2000 orientation data points were excluded from analysis. We obtained anatomy for 51 hΔC > SPARC flies. We quantified anatomy by counting cell body numbers near the FB. To compute the expression vector (Fig. S6B), we drew an ROI around the dorsal FB layer containing hΔC output tufts, split this ROI into 12 columns, converted normalized expression across these columns into a 360° polar plot, then converted the polar plot into a vector.

### Calcium imaging

For calcium imaging, we anaesthetized flies on ice and mounted them in a holder (https://ptweir.github.io/flyHolder/) using UV glue. We stabilized the head, proboscis, and body, removed the two front legs, and dissected the cuticle over the back of the head, removing trachea, airsacs, and muscles that covered the brain. The brain was bathed in extracellular saline (103 mM NaCl, 3 mM KCl, 5 mM TES, 8 mM trehalose dihydrate, 10 mM glucose, 26 mM NaHCO$_3$, 1 mM NaH$_2$PO$_4$H$_2$O, 1.5 mM CaCl$_2$2H$_2$O, and 4 mM MgCl$_2$6H$_2$O, pH 7.1–7.4, osmolarity 270–274 mOsm) bubbled with carbogen (5% CO$_2$, 95% O$_2$) during dissection and imaging.

2-photon imaging was performed using a pulsed infrared laser (Mai Tai DeepSea, Spectraphysics) at 920 nm with a Bergamo II microscope (Thorlabs), using a 20× water immersion objective (Olympus XLUMPLFLN 20×) and ThorImage 3.0 software. Power at the sample was 13–66 mW. Emitted photons were spectrally separated using red (tdTOM, 607/70 nm) and green (GCamp6f, 525/50 nm) bandpass filters and detected by GaAsP PMTs. Imaging area ranged from 47 × 47 μM to 122 × 74 μM and was identified using the tdTOM signal under epifluorescence. Images were acquired at ~5.0 frames per second. We excluded flies where we were unable to obtain 5 trials of each stimulus direction, 2 65C03 flies which showed rhythmic spike-like activity and did not respond to any phase of our stimulus, and one VT062617 fly in which no columns responded >2STD above baseline.

Wind and odor stimuli were delivered using a custom five-direction manifold[61] (MB052B, LH1396, 65C03-GAL4, 21D07, vFB split, and VT062617) or rotary union apparatus[21] (MB077B, MB082C, and 12D12). Charcoal-filtered air was passed through a flowmeter (Cole-Parmer) to direct air and odor at the fly from one of five directions using either proportional valves (manifold, EV-05-0905; Clippard Instruments) or solenoid valves (rotary union, LHDA1233115HA, Lee Company). Odor (apple cider vinegar, 10%) was diluted in distilled water on the day of the experiment. Airflow was ~25 cm/s for manifold experiments and 40 cm/s for rotary union experiments and maintained at a constant level throughout the experiment. Trials consisted of 5–10 s without stimuli, 10 s of wind alone, 10 s of odorized wind, 10 s of wind alone followed by 8–10 s of no stimulus. Odor/wind direction was randomized in blocks of five trials, with five blocks per fly (25 total trials). Imaging position, z-plane, gain, and power levels were adjusted as needed between blocks.

To extract time courses from calcium imaging data, we first used the CalmAn package[69] (Matlab) to implement the NoRMCorre rigid motion correction algorithm[70]. Motion correction was performed on the red (tdTOM) time series, then applied to the green (GCaMP6f) times series. ROIs were drawn by hand in ImageJ on a maximum intensity projection of the first trial of the tdTom time series, then manually adjusted for drift between trials. We targeted the following regions in each line. LH1396: dendritic processes in the LH; MB052B, MB082C and MB077B: putative axonal processes in the protocerebrum; FB5AB, 65C03, 12D12, and vFB split: FB layer innervated by each line. For hΔC imaging (VT062617) imaging, we drew ROIs across eight putative columns of the FB based on glomerular structure in the tdTom signal. ROIs were imported into MATLAB using ReadImageJROI[71]. ΔF/F was computed by dividing the green (GCaMP6f) time series by the average fluorescence of the baseline period (first 5 s of the trial, excluding the first sample due to shutter lag). Traces were resampled to 5 Hz if frame rate varied between experiments to compute means. Supplemental heat maps were normalized to maximum response across all trials within an individual fly.

### Electrophysiology

For whole cell patch clamp recordings, flies were mounted in a holder with hot wax and dissected. We used collagenase (5% in extracellular saline, Worthington Biochemical Corporation Collagenase Type 4) to remove the sheath over the brain. We visualized cell bodies using a 40× objective (Olympus, LUMPLFLN40XW), an LED source (Cairn Research MONOLED), and a filter (U-N19002 AT-GFP/F LP C164404), and cleaned them using extracellular saline and light positive pressure.

Glass pipettes (6–10 μOhms, World Precision Instruments 1B150F-3) were pulled using a Sutter P-1000 puller and filled with intracellular solution (140 mM KOH, 140 mM aspartic acid, 10 mM HEPES, 1 mM EGTA, 1 mM KCl, 4 mM MgATP, 0.5 mM Na3GFP, and 13 mM biocytin hydrazide). Current and voltage signals were amplified using either an A-M systems Model 2400 amplifier with

Brownlee Precision 410 preamplifier or a Molecular Devices Multiclamp 700B and digitized at 10 kz. We recorded from a total of 12 MBONs, 6 on each side that met our criteria of access:input ratio great than 5:1. Resting membrane potential ranged from −34.6 mV to −25.2 mV (mean −30.8 mV).

Stimuli for electrophysiology were delivered using simple olfactometer comprising a charcoal filter, flowmeter, and Teflon tube (4 mm OD, 2.5 mm ID) positioned <1 mm from the fly head on the right side. Solenoid valves (Lee company, LFAA1201610H) switched between humidified and odorized air. Odor pulses shown (10% apple cider vinegar) are 10 s long. Total airflow was 1.3 L/min. Membrane potential was extracted in Matlab by applying a 2.5 Hz low-pass Butterworth filter to raw voltage signals. Responses during the first 4 s of the odor period were compared to 2 s of baseline during wind only.

## Immunohistochemistry

Dissected brains were fixed for 15 min in 4% paraformaldehyde (in 1× PBS), washed 3× in PBS and stored at 4 °C until staining (within 2 weeks). To stain, fixed brains were incubated in 5% normal goat serum in PBST (1× PBS with 0.2% Triton-X) for 20–60 min, incubated overnight at room temperature in primary antibody (Table 1), washed 3× in PBST, incubated overnight in secondary antibody (Table 1), washed 3× in PBST then stored in PBS at 4 °C until imaging. Mounted brains were covered in vectashield (Vector Labs H-1000), sealed with coverslips, and imaged using a 20× objective (Zeiss W Plan-Apochromat 20×/1.0 DIC CG 0.17 M27 75 mm) on a Zeiss LSM 800 confocal microscope at 1–1.25 μM depth resolution. Images presented are maximum z-projections over relevant depths.

We dissected, stained, and imaged the following number of brains to determine expression patterns: Fig. 3a: MB052B > transtango $N = 2$, LH1396 > transtango $N = 3$, Fig. 4A: FB5AB (21D07||CLIN) > Chrimson $N = 3$, 65C03 > Chrimson $N = 1$, 12D12 > Chrimson $N = 2$, vFB split> Chrimson $N = 1$. Figure 6A: hΔC > SPARC $N = 51$, Fig. 6E: hΔC split> Chrimson $N = 1$, Fig. S4B: 21D07||ChAT $N = 2$, 21D07 GABA $N = 2$, 65C03||ChAT $N = 1$, 65C03||GAD1 $N = 1$, 12D12||ChAT $N = 1$, 12D12||GAD1 $N = 1$, vFBsplit||ChAT $N = 3$ Fig. S5A: VT062617||ChAT $N = 2$, VT062617|| GAD1 $N = 1$. For Chrimson experiments, we dissected females from the same cross as experimental males.

## Connectomic analysis

Data from the hemibrain connectome[22] were obtained from neuprint explorer (https://neuprint.janelia.org, hemibrain:v1.1) and analyzed using MATLAB and Python (see Code availability). For the analysis shown in Fig. 5B, the location of FB5AB synapses onto hΔC was determined by their x-position, with no filtering or constraints. For the analysis in Fig. 4B, we counted all FB tangential neurons two synapses downstream of the following projection neurons: VM7d_adPN, VM7v_adPN, DM1_lPN, DM4_adPN, DM4_vPN, VA2_adPN, DP1l_adPN, DP1l_vPN, DL2d_adPN, DL2d_vPN, DL2v_adPN, DC4_adPN, DC4_vPN, DP1m_adPN, DP1m_vPN in which the intermediate neurons contained LH (they were lateral horn neurons) and where synaptic weights exceeded a weight of 3.

## Modeling

The FB steering circuit model implemented in this study contains three types of inputs—heading, wind, and odor; two intermediate layers corresponding to hΔC and mutual inhibition local neurons; and an output layer corresponding to left PFL3, right PFL3, and PFL2 neurons, which modulate angular velocity and walking speed. Heading inputs go directly to PFL3 and PFL2, while odor and wind inputs are integrated by hΔC neurons, processed by the mutual inhibition layer, and then integrated by PFL3 and PFL2. Angular velocity depends on the relative activity of left and right PFL3, while speed depends on the activity of

### Table 1 | Antibody sources and dilutions

| Antibody | Source | Dilution | Identifier |
|---|---|---|---|
| Chicken anti-GFP | Fisher Scientific | 1:50 | RRID:AB_1074893 |
| Mouse anti-nc82 | DSHB | 1:50 | RRID:AB_2314866 |
| Rabbit anti-DsRed | Clontech | 1:500 | 632496 |
| Rabbit anti-GABA | Sigma | 1:100 | RRID:AB_477652 |
| Alexa488-conjugated goat anti-chicken | Fisher Scientific | 1:250 | RRID:AB_2534096 |
| Alexa633-conjugated goat anti-mouse | Fisher Scientific | 1:250 | RRID:AB_2535719 |
| Alexa568-conjugated goat anti-rabbit | Fisher Scientific | 1:250 | RRID:AB_2315774 |
| Alexa568-conjugated streptavidin | Fisher Scientific | 1:1000 | RRID:AB_2576217 |

PFL2. All modeling was performed in Matlab (see Code availability). Model parameters are given in Table 2. All dynamic systems were computationally approximated using Euler's method of integration with a time step of 0.05 s.

**Input layer: heading.** Heading input to PFL3 and PFL2 was simulated by phase-shifting heading-tuned one-cycle sinusoids:

$$\text{heading\_input} = \cos(\text{FB}_{\text{location}} - (180° - \text{heading} + \text{phase\_shift})) + 1 \tag{1}$$

where $\text{FB}_{\text{location}}$ refers to the spatial location within the FB ($0° = $ left FB, $360° = $ right FB), heading refers to the heading direction of the fly, and phase_shift refers to the anatomical shift introduced by PFL neurons entering the FB. The phase shift parameters were taken directly from the connectome[31] and were defined as the following: phase_shift$_{\text{left\_PFL3}} = -90°$, phase_shift$_{\text{right\_PFL3}} = +90°$, phase_shift$_{\text{PFL2}} = +180$.

**Input layer: wind.** Wind-tuned bumps of activity in hΔC neurons were simulated using one-cycle sinusoids that depended on the encoding strategy. For allocentric representation (Fig. 7), the bump of activity in hΔC neurons follows the allocentric direction of wind and was defined by the following:

$$\text{wind\_input}_{\text{allocentric}} = \cos(\text{FB}_{\text{location}} - (180° - \text{wind\_direction}_{\text{allocentric}})) + 1 \tag{2}$$

where wind_direction refers to the allocentric wind direction. For frontal representation (Fig. S7) hΔC wind activity was constructed by phase shifting and summing the activity of a left and right population of wind-sensitive PFNs[21], each displaying a heading-tuned bump scaled by wind direction:

$$\text{wind\_input}_{\text{frontal}} = \cos(\text{FB}_{\text{location}} - (180° - \text{heading} + \text{phase\_shift}_{\text{left}})) * \text{scale}_{\text{left}}$$
$$+ \cos(\text{FB}_{\text{location}} - (180° - \text{heading} + \text{phase\_shift}_{\text{right}})) * \text{scale}_{\text{right}} \tag{3}$$

where phase_shift refers to the anatomical shift introduced by PFN neurons entering the FB, and scale refers to the wind-tuned scaling of bump amplitude. The phase shifts differed for the two populations of PFNs and were defined as: phase_shift$_{\text{left}} = +45°$, phase_shift$_{\text{right}} = -45$[31] The scaling corresponded to the wind tuning of the PFNs, which differed between the left and right PFN populations[21]

$$\text{scale} = \sin(\text{wind\_direction}_{\text{egocentric}} + (90° - \text{peak})) \tag{4}$$

where wind_direction$_{\text{egocentric}}$ is the egocentric wind direction and peak is the egocentric wind direction that maximally activates the PFN cell type (peak$_{\text{left}} = -45°$, peak$_{\text{right}} = +45°$).

**Table 2 | Values of steering circuit model parameters with their units, roles, and sources listed**

| Parameter | Value | Units | Role | Source |
|---|---|---|---|---|
| phase_shift$_{left\_PFL3}$ | −90 | deg | Anatomical shift of left PFL3 neurons entering FB | [31] |
| phase_shift$_{right\_PFL3}$ | 90 | deg | Anatomical shift of right PFL3 neurons entering FB | [31] |
| phase_shift$_{PFL2}$ | 180 | deg | Anatomical shift of PFL2 neurons entering FB | [31] |
| phase_shift$_{left\_PFN}$ | 45 | deg | Anatomical shift of left PFN neurons entering FB | [31] |
| phase_shift$_{right\_PFN}$ | −45 | deg | Anatomical shift of right PFN neurons entering FB | [31] |
| peak$_{left}$ | −45 | deg | Peak wind direction tuning of left PFNs | [21] |
| peak$_{right}$ | 45 | deg | Peak wind direction tuning of right PFNs | [21] |
| optogenetic_power | Low: 0.75, High: 2.3 | Activity | Activation of hΔC neurons by optogenetic stimulus | Optimized to match behavior |
| $k_{hΔC}$ | 0.1 | – | Slope of hΔC activation function | [72] |
| $θ_{hΔC}$ | 0.7 | Input | Threshold of hΔC activation function | Optimized to match behavior |
| tau$_n$ | 10 | ms | Time constant for exponentially filtered white noise | [72] |
| $σ_n$ | 0.03 | Activity | Standard deviation of exponentially filtered white noise | [72] |
| $W_{ii}$ | I <-> i+4: 1.1 Otherwise: 0 | – | Synaptic strength between mutually inhibited neurons | [72] |
| $g$ | 0.5 | – | Adaptation weight for mutual inhibition neurons | [72] |
| tau$_u$ | 1 | ms | Time constant for mutual inhibition neurons | [72] |
| tau$_a$ | 100 | ms | Time constant for slow feedback adaptation | [72] |
| $k_{mi}$ | 0.1 | – | Slope of mutual inhibition activation function | [72] |
| $θ_{mi}$ | −0.15 | Input | Threshold of mutual inhibition activation function | Adjusted from[72] |
| tau$_{pfl}$ | 1 | ms | Time constant for PFL3 and PFL2 neurons | [72] |
| $k_{pfl}$ | 0.1 | – | Slope of PFL activation function | [72] |
| $θ_{pfl}$ | 1.7 | Input | Threshold of PFL activation function | Optimized to match behavior |
| $λ$ | 0.0003 | Rate | Baseline turn rate for 20000 Hz simulated data | [10] |
| $σ_{turn}$ | 20 | deg/s | Standard deviation of angular velocity distribution | [10] |
| $m1$ | 0.03 | deg/(s*activity) | Constant coupling PFL3 activity to angular velocity at 20,000 Hz | Optimized to match behavior |
| $v_{base}$ | 6 | mm/s | Baseline groundspeed | [10] |
| $m2$ | 0.25 | mm/(s*activity) | Constant coupling PFL2 activity to groundspeed | Optimized to match behavior |

Note: many parameters were manually fit to match simulation data with behavioral trajectories. While not explored in this study, many of these parameters (e.g., activation thresholds) are flexible and can be adjusted together to keep the steering circuit functioning across a range of sinusoid amplitudes.

**Intermediate layer 1: hΔC neurons.** The total activity of hΔC neurons was calculated as the sum of wind input and optogenetic input:

$$\text{hΔC\_input} = \text{wind\_input} * \text{wind\_gain} + \text{optogenetic\_input} \quad (5)$$

where wind_gain is a binary variable set to 1 only if odor is present and optogenetic_input is a vector of sparse or broad optogenetic patterns. Sparse optogenetic patterns were generated by creating a 20-element vector, with each element stochastically set to 0 or 2.3 at a 15% probability. Broad optogenetic patterns were generated by creating a 20-element vector with every value set to the same value (low light = 0.75, high light = 2.3). These values were chosen to best match simulated trajectories with real behavior.

The output of the hΔC population was calculated by phase-shifting the hΔC activity by 180°. To simplify the pathways connecting hΔC neurons with mutual inhibition circuits, PFL3 neurons, and PFL2 neurons, which all involve at least one additional cell type, the output of hΔC neurons was transformed from 20-neuron space into 8-column space and then thresholded using a sigmoidal activation function:

$$\text{hΔC}_{output} = S_{hΔC}\left(\mathbf{W_{hΔC}} * \phi\left(\text{hΔC}_{activity}\right)\right) + \boldsymbol{n} \quad (6)$$

where $S_{hΔC}$ is the hΔC sigmoid activation function with slope $k_{hΔC}$ and threshold $θ_{hΔC}$, $\mathbf{W_{hΔC}}$ is the feedforward transformation matrix, $\phi$ is the 180° phase shift function, and $\boldsymbol{n}$ is exponentially filtered white

noise. The matrix used for the space transformation was inspired by the connectivity between local neurons in the fan-shaped body. To construct this matrix, each neuron's output was set as the size of one column, such that there were 20 overlapping column-sized outputs. The matrix element for column$_i$ and neuron$_j$ was then set as the amount of overlap between column$_i$ and the output of neuron$_j$, and the total input to column$_i$ from all neurons was normalized to equal 1. The exact structure of $\mathbf{W_{hΔC}}$ was not critical, as long as the matrix was normalized and maintained the approximate location of wind sinusoids in FB space. The filtered white noise was defined by the following equation[72]

$$\frac{d\boldsymbol{n}}{d\boldsymbol{t}} = \frac{-\boldsymbol{n}}{\text{tau}_n} + σ_n * \sqrt{\frac{2}{\text{tau}_n}} * η(t) \quad (7)$$

where $n$ is the noise variable, tau$_n$ is the noise time constant, $σ_n$ is the noise standard deviation, and $η(t)$ is white noise with zero mean and unit variance.

**Intermediate layer 2: mutual inhibition local neurons.** hΔC columnar output was relayed into an inhibitory recurrent network, consisting of eight inhibitory neurons with mutual connections between neurons located in opposite columns. Neurons in this network also contained a slow adaptation parameter, $\boldsymbol{a}$, that curtailed input when neural activity was high. This system was described using the following

differential equations:

$$\mathrm{tau}_u * \frac{du}{dt} = -u + S_{mi}\left(-\mathbf{W_{ii}} * u - g * a + h\Delta C_{output}\right) \tag{8}$$

$$\mathrm{tau}_a * \frac{da}{dt} = -a + u \tag{9}$$

Where $u$ is an 8-element vector corresponding to the activity of each inhibitory neuron, $a$ is an 8-element vector corresponding to the adaptation of each inhibitory neuron, $\mathbf{W_{ii}}$ is the recurrent weight matrix connecting the inhibitory neurons, $g$ is the adaptation weight, $h\Delta C_{output}$ is the columnar output of the h$\Delta$C neurons, $\mathrm{tau}_u$ is the time constant for neural activity, $\mathrm{tau}_a$ is the time constant for adaptation, and $S_{mi}$ is the mutual inhibition sigmoid activation function with slope $k_{mi}$ and threshold $\theta_{mi}$. The equations and parameters used to model the mutual inhibition circuit were taken from ref. 72.

The output of the mutual inhibition layer was calculated by phase-shifting $u$ by 180°. This output was subtracted from the h$\Delta$C output to create the total local output sent to PFL3 and PFL2.

**Output layer: PFL2 and PFL3 neurons.** The activity of left PFL3, right PFL3, and PFL2 were calculated by summing the excitatory output of h$\Delta$C neurons, inhibitory output of the mutual inhibition layer, and phase-shifted excitatory output of the heading system, and then converting this input into activity using the following dynamic system:

$$\mathrm{tau}_{pfl} * \frac{dpfl}{dt} = -pfl + S_{pfl}(input) \tag{10}$$

where $\mathrm{tau}_{pfl}$ is the time constant for PFL neurons and $S_{pfl}$ is the PFL sigmoid activation function with slope $k_{pfl}$ and threshold $\theta_{pfl}$.

**Output layer: navigation modulation.** Turning was calculated by comparing the activity of right PFL3 neurons and left PFL3 neurons and adding a noise term:

$$\triangle\text{heading} = m1 * \left(\sum \mathbf{PFL3_{right}} - \sum \mathbf{PFL3_{left}}\right) + p(t) * G \tag{11}$$

where $m1$ is the coupling constant between PFL3 activity and turning, $p(t)$ is a binary Poisson variable with $\lambda$ rate, and G are angular velocity values drawn from a Gaussian distribution with zero mean and $\sigma_{turn}$ variance. The random turn equations and parameters were taken from ref. 10 while the coupling constant was selected to best match behavioral trajectories.

Groundspeed was calculated by summing the activity of PFL2 neurons according to the following equation:

$$v = v_{base} + m2 * \sum \mathbf{PFL2} \tag{12}$$

where $v_{base}$ is the baseline speed and $m2$ is the coupling constant between PFL2 activity and speed. The baseline speed was taken from ref. 10 while the coupling constant was selected to best match behavioral trajectories.

**Output layer: simulated trajectories.** The model simulates heading and groundspeed over time in response to wind/odor or optogenetic stimuli. These values were converted into movement in $x$ and $y$ using:

$$\begin{aligned} x_{n+1} &= x_n + \triangle t * v_n * \sin(\text{heading}_n) \\ y_{n+1} &= y_n + \triangle t * v_n * \cos(\text{heading}_n) \end{aligned} \tag{13}$$

where $\triangle t$ is the time step between data points, $v_n$ is the current groundspeed, and $\text{heading}_n$ is the current heading.

**Other.** The sigmoid activation function used for h$\Delta$C neurons, the mutual inhibition circuit, and PFL neurons was the following.

$$\text{activity} = \frac{1}{\left(1 + e^{\frac{-(x-\theta)}{k}}\right)} \tag{14}$$

where $x$ is the input, $\theta$ is the activation threshold, and $k$ is the activation slope.

**Quantification and statistical analysis**

**Behavior.** Based on previous behavior data[10] we assumed these data were not normally distributed and applied non-parametric statistics. We used the two-sided Wilcoxon signed rank test (MATLAB signrank) to compare the average baseline value of each parameter to the average of the parameter over a window of interest. Between genotype comparisons (genetic silencing experiments) were made between baseline subtracted parameter values using the two-sided Mann–Whitney $U$ test (MATLAB ranksum). For behavioral time courses we display standard error around the mean. For summary plots we present standard deviation around the mean. Bonferroni corrections for multiple comparisons were applied based on the number of genotypes tested labeling similar neuron types (i.e. 6 in Fig. 1D, 21 for all dorsal FB inputs in Fig. 3B, C) or on the number of comparisons made to control in genetic silencing experiments (3 in Fig. 1F).

**Imaging and physiology.** We used parametric tests for calcium imaging and electrophysiological data. When testing for significant differences in odor responses across directions we averaged individual trials across flies, and performed a one-way ANOVA across pooled trials across flies (MATLAB anova1). When assessing differences between wind ON responses and odor responses we calculated mean wind and odor responses in 10 s windows following ON. We compared wind and odor periods using a two-tailed paired student's $t$-test (MATLAB ttest, Fig. S4E, Fig. S5A). Similarly, we performed two-tailed paired student's $t$-test when comparing wind and odor responses in MBON electrophysiology experiments and a two-tailed unpaired student's $t$-test (MATLAB ttest2) when comparing odor responses between ipsilateral and contralateral MBONs. To assess the decay of fluorescence response to odor over time (Fig. 4D), we computed the average response to the odor presentation period across flies for five trial blocks, including one trial of each direction. We averaged the response across all flies imaged, and normalized by the mean response in trial block one. Shaded regions depict standard error across flies around the mean.

**Classification training.** We trained classifiers to assess if an observer could correctly identify whether wind was coming from the left, right or front relative to the fly based on calcium imaging data from select neuron groups. We pooled the data for 45 and 90 degrees left and right for this analysis. We trained classification tree models on calcium response data during odor ON (5 s, Figs. 2F, 4D) and during wind ON and OFF (5 s, Supplementary Figs. 2H, 4E). Model were trained using two predictors: the fly identity, and the Z-score of the calcium response during the specified window. We took the Z-score of the response to standardize the data between electrophysiology firing rates (PFNa) and calcium responses (all other neurons). As we recorded both left and right LNa neurons simultaneously we asked how well the difference between the activity of the two performed. We trained classification trees (Matlab's fictree) to identify wind direction and tested model performance using 10-fold cross validation error

(Matlab's KfoldLoss). We compared this error to classification trees that were trained on the same data with shuffled labels, with 50 random iterations of shuffled labels.

We also designed classifier to assess if an observer could distinguish between wind ON activity and odor ON activity. We used the same parameters in the direction classifier but pooled all data across all wind directions. We performed the same analysis to assess if an observer could distinguish between baseline activity and odor ON activity in Supplementary Figs. 2F and 4F.

### Reporting summary
Further information on research design is available in the Nature Research Reporting Summary linked to this article.

## Data availability
Behavior and imaging data generated in this study have been deposited on Zenodo at https://zenodo.org/record/6863832#.Ytlati-B3UJ, DOI: 10.5281/zenodo.6863832. The connectomic data used in this study (hemibrain version 1.1) are available on neuprint https://neuprint.janelia.org/. Source data are provided with this paper. Any additional information required to reanalyze the data reported in this paper is available from the Lead Contact upon request.

## Code availability
All original code is available on Github at https://github.com/nagellab/Mathesonetal2022. Model code is available on Zenodo at https://zenodo.org/record/6762105#.YtlbYi-B3UJ, https://doi.org/10.5281/zenodo.6762105.

## Materials availability
No new transgenes were created for this study. Transgenic stocks are available on request from the Lead Contact.

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

## Acknowledgements

The authors would like to thank Marc Gershow, Karla Kaun, Tom Clandinin, Calude Desplan, Michael Dickinson, Michael Reiser, Matthew Clark, Matthieu Louis, Richard Mann, Gilad Barnea, Rachel Wilson, Tzumin Lee, Janelia Flylight, and the Bloomington and Vienna *Drosophila* Resource Centers for fly stocks. We thank Michael Long, Elizabeth Hong, Floris van Breugel, David Schoppik, Gaby Maimon, and members of the Nagel and Schoppik labs for helpful input on the manuscript. We thank Haim Schoppik for help with data archiving. This work was supported by R01DC017979, RF1NS127129, NSF Ideaslab (IOS-1555933) and NSF Neuronex grants (NSF 2014217), as well as a McKnight Scholar Award to K.I.N. M.H.S. was supported by an NSF CAREER award IOS-204720. T.A.C. was supported by a Dean's Dissertation Fellowship from NYU. A.M.M. was supported by a Diversity Supplement to R01DC017979 and a McKnight Pecot Fellowship.

## Author contributions

K.I.N., A.M.M.M., and A.J.L. conceived the project. A.M.M.M. performed the majority of behavioral, imaging, electrophysiology, and anatomy experiments, and generated new genetic stocks. A.J.L. performed SPARC experiments with A.M.M., performed connectomic analysis, and developed the computational model with input from K.I.N and A.M.M.M. A.M.L. contributed to behavioral experiments. T.A.C. performed a subset of electrophysiology experiments. M.H.S. provided the genetic strategy to isolate CX neurons. K.I.N., A.M.M.M., and A.J.L. wrote the paper with input from the other authors.

## Competing interests

The authors declare no competing interests.

 
