## [Peer review file · Nature Communications]

REVIEWER COMMENTS

Reviewer #1 (Remarks to the Author):

Summary

In this paper Matheson et al show that non-directional information about odors is combined with directional information about wind direction in the fan shaped body (FB) of the central complex (CX) in flies. This paper delivers first data on a central hypotheses generated by the fruit fly connectome, namely that contextual information is combined with spatial information in the CX. This is highly important work and it is carried out in an exemplary manner. A wide range of methods is used to systematically narrow down the possible integration sites of directional and non directional information from the antennal lobes, to the mushroom body, lateral horn, and input neurons to the FB, to finally converge on specific interneurons of the FB, the hDeltaC cells. Using connectomics data from the recently published hemibrain connectome, the authors generate an hypothetical circuit that is consistent with their data and that yields a network that can generate upwind orientation in flies in the presence of an attractive odor, while downwind orientation without an odor. Such context dependent switches in behavior are an important principle of how brains make decisions in general, and illustrate how sensory information is used differently depending on learned or innate context.

While I am not a fly geneticist and thus cannot comment on the details of fly lines and potential pitfalls, potentially missing controls etc, the behavioral, physiological, and imaging work is of very high quality, presented clearly and in a transparent way.

My only concern is that the final model of the paper is presented as an actual neural circuit, rather than a conceptual model that can be used as a starting point for further studies. The presented circuit, as illustrated in figure 6, is only partially possible when recapitulating it in the fly connectome data. Firmly established connections need to be more clearly distinguished from idealized and simplified connections, especially if they are crucial to the central conclusions of the paper.

That said, the paper is a beautiful and important paper, well written and enjoyable to read. It provides an important contribution to the field and will serve as starting point for many lines of investigations that aim at linking neural circuits to context dependent behavioral choices in animal brains.

Major comments

1) Stimulation of the hDeltaC neurons leads to turns. While such a response is expected when stimulating the non directional neurons representing valence (their information is downstream combined with directional wind information), I do not clearly see how activation of the entire population of hDeltaC neurons would lead to directed turns. This is because in these neurons wind information is already supposed to be combined with odor information and only those columns that receive PFN input as well as odor input will be naturally activated when initiating a turn into the appropriate direction (if I understood the model correctly). If all these cells are activated at the same time, no obvious source of asymmetry would be present and steering-neurons on both sides of the brain would be activated. Maybe the authors could clarify what the patterns of activation in hDeltaC neurons look like when driving these cells artificially. How is turning in response to global activation of these cells compatible with the model presented later in the paper?

2) The interpretation of the data and the construction of a circuit to drive navigation appears to not be fully supported by the connectomics data. While elegant in its formulation and easy to understand, the links between the involved neurons, in particular between PFNa, hDeltaC and PFL3 neurons are not very strong (or do not exist) and hence the conceptual model presented is an idealized oversimplification. As the model is consistent with the presented data (but see point 1 above), I do believe it yields a valid hypothesis that can serve as a starting point for future studies, but the authors need to rework the presentation in a way that allows the readers to distinguish established neural connections from simplifications. As it stands, the reader is left with the impression that the diagram in Figure 6A depicts an actual and complete circuit, which is not backed up by sufficient connectomics data. In particular I found the following contradictions with the hemibrain connectome (Hulse et al 2021, mostly based on figure 33 and figure 33 supplement 1 of that paper):

- PFNa neurons (most strongly responding to wind input) have no direct connection to PFL3 (or PFN2) neurons.

- They also have no strong indirect connection to PFL3 neurons, but form a separate pathway towards PFL1 neurons. While a weak connection to hDeltaC exists, and these neurons are connected to PFL3 via FC2B cells, these connections are weak and do not present an obvious major pathway through the FB. At this level of connectivity, many potential pathways through the FB exist, especially across three sequential synapses.

- PFNp and PFNm cells are a heterogeneous population of neurons with many different subtypes that connect to different targets. Grouping them with one another and with the very different PFNa cells is a bit misleading and it should at least be mentioned that this group of cells can lead to multiple different pathways. PFNm neurons also have no direct connections to PFL3 cells.

- The only direct pathway between PFN and PFL3 cells is realized by PFNp neurons, yet only one of these types is connected to hDeltaC neurons: PFNp_c. Hence the direct and indirect pathway from figure 6A

can only exist for PFNp_c (if the same cell type should serve both pathways). Is anything known if these cells respond to wind stimuli?

In summary, the hypothesized direct and indirect pathways exist only if PFN neurons are treated as a group, rather than the diverse set of cell types they are. Only a single cell type (PFNp_c) can serve both pathways equally, but this is not the neuron with the strongest wind response (if I remember this correctly) and the pathway does not stand out anatomically. If, alternatively, different neurons feed into the direct and indirect pathways, a nicely balanced model (as presented) is not easy to achieve. Whether such a pathway is still feasible could be shown by computational models that take into account the actual synaptic weights of the connections (in future work).

I suggest to clarify and rework the model part of the paper, so that it becomes more clear that it is a hypothesis that illustrates an important and novel concept, rather than an actually existing circuit.

Minor comments

line 130: unclear what is meant by 'offset'. The term offset is used in the CX literature to describe an initial phase difference of the head direction code between individual flies, but that is clearly not what is meant here.

line 380: Is the assumption that the PFL3 neurons are inhibitory based on any data? If not, then this should be stated more clearly.

line 476: Cx should be CX

(Stanley Heinze)

Reviewer #2 (Remarks to the Author):

Although it has been known that animals use both odor and wind direction cues to navigate the olfactory environment, the underlying neural circuit mechanisms are poorly understood. This

manuscript utilizes the strength of various technologies available in flies including quantitative behavioral analysis, in vivo physiology, optogenetic manipulation, EM level connectome analysis, computational modeling, and identifies a neuron type, hDeltaC neuron, that integrates the information about odor and wind direction. Because the upstream neurons mainly convey either odor or wind direction information, the authors conclude that hDeltaC neuron is a site of integration. However, some data do not support their claims and there are several substantial issues that need to be addressed.

Major comments

1. One of the major claims is that wind-direction tuned activity of hDeltaC neuron is gated on by an odor. However, although hDeltaC neuron responds more weakly to wind onset than to an odor (Fig. 5G), it responds more strongly to wind offset than to an odor for several wind directions, at least for the example neuron shown in Fig. S5A, suggesting that it can transmit information about changes in wind speed and direction without gating by an odor. Quantification of data for all the recorded cells and re-evaluation of the claim are warranted.
2. Similarly, LH1396 neurons (Fig. S2E) and FB5AB neurons (Fig. S4D) seem to show distinct wind tuning at wind offset and onset, respectively. Therefore, although these cells may not exhibit significant directional tuning at the transition of wind and wind+odor, they are tuned to wind direction at wind onset and offset. Based on to the way the authors previously characterized PFN neurons as wind-encoding (Currier et al., eLife, 2020), these neurons should also be described as wind-encoding besides odor-encoding.
3. In Fig. 5H, the responses of hDeltaC neurons are re-aligned to the maximally responsive direction, but the original data without alignment is useful to examine if there is a relationship between the column and the maximally responsive direction. If this relationship is different across flies like the relationship between wind direction and peak response (Fig. 5E and line 307-308), what additional mechanisms should be required to explain the contribution of these cells to consistent upwind navigation of the flies?
4. Fig. 5I is meant to show the causal effect of hDeltaC neurons on the upwind navigation. However, based on the physiology as well as the circuit model built in Fig. 6, this experiment is problematic because optogenetic stimulation activates all the hDeltaC neurons regardless of the fluctuating wind direction that the fly is experiencing at each moment during the light application period. Accordingly, the outcome is a turning behavior that is hard to explain. A proper approach is local activation of certain columns using focal P2X2 activation (e.g. Fujiwara et al., Nat Neurosci, 2016; Green et al., Nature, 2017) or optogenetic Chrimson activation (e.g. Kim et al., Science 2017), that require a fly on the ball. Even though the entire hDeltaC neurons are manipulated, an alternative is inhibition (e.g. GtACR) experiment in the freely behaving arena where the loss of upwind navigation would indicate the necessity of these neurons.
5. In the model shown in Fig. 6, I believe it would be easier to follow the logic if the implicit details or neurons are added to the circuit. For example, how can panel A generate asymmetrical activation of right and left PFL3? Who are the integrators of left PFN and right PFN (Fig. 6 and line 364)?

Minor comments

6. A tuning index (the difference between response to stimuli from 90 deg and -90 deg) does not account the responses to stimuli from the other three directions. There are better indices that quantify the direction selectivity.
7. Line 309. Fig. S3B should be Fig. S5A.
8. Line 73 Huoviala et al., 2020 and line 187 Schulze et al. 2015 are not listed in the References.
9. Line 1313. Jung et al., 2015 reported the ACV-responses of olfactory receptor neurons and not projection neurons.

Reviewer #3 (Remarks to the Author):

This a very interesting and convincing account of how odour signals reach the steering centres of the insect brain (the Central Complex, CX) and are integrated with directional signals from wind to drive behaviour. It nicely combines behavioural, anatomical, optogenetic (silencing and activation) and modelling studies to present an unusually complete picture, and will be of substantial significance in the field and beyond.

Specifically they: demonstrate optogenetic activation of receptor neurons can substitute for odours in promoting upwind navigation behaviour, and these neurons are required; show a subset of mushroom body and lateral horn neurons produce movement relative to the wind, and are responsive to odour but not wind direction; identify through screening tangential neurons in the fan-shaped body, downstream from the MB, that promote upwind behaviour and respond to odour; used the connectome to identify FB local neurons that receive inputs from these tangential cells and previously identified columnar neurons that respond to wind direction; use calcium imaging to characterise the response of these neurons and optogenetic activation to show they are involved in steering; and finally modelling to provide a plausible interpretation of the circuit function.

The following points of criticism are suggestions to improve the clarity of presentation:

Abstract final sentence (and elsewhere) "spatial and non-spatial information" - I feel throughout the manuscript, "directional and non-directional" might be a more accurate terminology. E.g. the odour events experienced by an animal moving in and out of an odour plume are spatial information.

Introduction

line 50: it is not obvious how vision can provide information about the prevailing wind. Is the idea that optic flow caused by the wind disturbing motion can be interpreted as wind direction? This is far from straightforward, requiring disambiguation from self-motion etc. It might be simpler to eliminate 'or vision' and focus on mechanosensation, introducing the potential use of vision more clearly in the discussion.

line 53 "where these two types of information are integrated is not known for any species" seems too strong, e.g., see Sánchez-Alcañiz, J. A., & Benton, R. (2017). Multisensory neural integration of chemical and mechanical signals. *Bioessays*, 39(8), 1700060.

line 56 "precise roles are not clear" - it is arguable that the 'roles' for these neuropils in some particular behaviours have been shown at least as 'precisely' as is shown for the behaviour examined here. This introduction should also mention how the lateral accessory lobe has been closely implicated in odour steering behaviour in moths.

Results

It might be clearer to use 'vinegar' as the short-hand for 'apple cider vinegar' rather than ACV.

Line 146 "activation in absence of wind...produced offset search" - is there information on the effect of vinegar in the absence of wind, or is this not possible to control precisely enough to reveal a similar effect? It is an interesting observation that could be better followed up (e.g. in the model) that upwind orientation and offset search might be distinct behaviours, but it is not clear why these data suggest that "these two behaviours are driven by distinct but overlapping populations of olfactory glomeruli" e.g. it could be an effect of the different temporal dynamics of neural responses to vinegar and optogenetic activation.

Line 163, This section might be better if the distinction of odour gradient behaviour, involving reduced curvature, and odour behaviour in flow is introduced at the start.

Line 212 on, it is a little unclear why "to test specifically whether alpha³ MBONs show wind direction tuning", the experiments did not use the same presentation of wind from different directions as the

previous tests, but instead looked at the differential response of ipsilateral and contralateral neurons to an odour presented from one side. The figure ref here is incorrect (2F->2G)

line 299 "might gate synaptic output" - it seems quite important to be clear about the envisioned axo-axonic mechanism here (it is touched on in the discussion). 'Gating' of output is rather different to the 'integration' described in fig 5. What might be understood from the inconsistent wind direction tuning? line 315, it seems unclear whether h-delta-cell activation produces specifically wind-directed turning or just turning?

Model: it is potentially confusing that the illustration in 6E assumes that 'heading bump' of the fly (from EPGs) is oriented in the same direction as the wind. It is consequently not obvious that the "+45 degrees and -45 degrees" are anatomical shifts of this bump (which could be in any direction relative to the wind) by one column due to PFN anatomy, which is then subsequently modulated in amplitude by the wind direction. In 6E the two 'bumps' seem to be at +/-45 degrees relative to the FBN and I initially assumed they were 'fixed' in this way and had to read the model details to understand otherwise. In fact I am still unsure - is the PFN bump (before amplitude modulation) determined by the wind direction? In the model description line 818, the 'heading' of the fly produces the bump, but can this be arbitrary with respect to the wind? Can the bump itself have an arbitrary (but consistent) offset from the heading (as observed in flies)? Does the model still work under these conditions?

Figure 6F also seems a bit misleading, showing what looks like an all-or-nothing amplitude modulation rather than clearly illustrating how the preferred tuning of the TN cells is assumed to modulate the amplitude.

The general story of the paper led me to expect that the model would illustrate the smooth integration of the spatial and non-spatial information, e.g. with a simulated agent exposed to wind and changing its behaviour in the presence of odour. But the model results shown are more static, just the independent steering output of the two putative pathways, with the assumption (not modelled) that the presence of odour switches the system into use of the indirect pathway (and switches off the direct pathway?). The model itself does not include any term for odour, or how the h-delta-C neurons integrate (or are gated) by it, or how (line 336) "competing pathways" "alter the balance of activity between" them based on odour.

It would be appropriate here or in the discussion to reference this paper:

Goulard R, Buehlmann C, Niven JE, Graham P, Webb B (2021) A unified mechanism for innate and learned visual landmark guidance in the insect central complex. *PLOS Computational Biology* 17(9): e1009383. <https://doi.org/10.1371/journal.pcbi.1009383>

- which provides a similar, but more subtle model for how a non-directional 'value' signal from the MB could be integrated with directional steering in the CX through PFN/PFL interaction.

Discussion

A good point is made about the separation of identity and directional information but seems easily tested whether learning of value in one behavioural task, e.g. walking or proboscis extension, generalises to another, e.g. flight choice decisions. Is there not any citable study of this?

Around line 530, a plausible possibility is that, compared to direct steering pathways, the CX allows the animal to maintain steering in a particular direction even when the cue or stimulus for steering is intermittent. The Goulard et al. (2021) model mentioned above demonstrates both this and the ability to adopt a course at a particular angle to the directional cue (line 535).

NCOMMS-21-43642: Response to Reviewers

We would like to thank the reviewers for their thoughtful and constructive feedback on our manuscript. We were delighted by the positive feedback from most reviewers, including Reviewer 1's comment that it is "a beautiful and important paper... an important contribution to the field and will serve as starting point for many lines of investigations" and Reviewer 3's comment that the paper "will be of substantial significance in the field and beyond." We also appreciate Reviewer 2's concerns which have pushed us to more carefully quantify our claims. As detailed below, we have revised multiple aspects of the manuscript. All new text is marked in blue in the revised manuscript and supplement. We look forward to hearing the reviewers' appraisal of our revised manuscript.

Major revisions:

1) Role of h Δ C neurons in navigation

All three reviewers raised concerns with the h Δ C activation experiment. In our first submission, we reported effects of broadly activating all h Δ C neurons, while our imaging data suggests that only a small subset of these neurons are active at one time *in vivo*. To address this concern, we have performed sparse activation of h Δ C neurons in 60 flies using the sparse expression tool SPARC2 (Isaacman-Beck et al. 2020) as well as control experiments in 34 flies. For experimental flies, we dissected and stained every fly to determine which columns were labeled. Using this paradigm, we found that sparse activation in many flies causes them to turn and walk in an arbitrary but reproducible direction, unique to each fly (new Figure 6). This led to orientation histograms that were strongly peaked. In contrast, we did not observe this behavior in flies where the SPARC construct was driven with an empty-Gal4 driver. We interpret this result to mean that asymmetric activation of h Δ C neurons can encode a "goal heading" that causes the fly to persistently walk in a particular direction. We also provide a more extensive dataset showing that uniform activation of h Δ C neurons drives unstable re-orientation behavior. Finally, we show using acute silencing that h Δ C activity is required for persistent upwind orientation throughout the odor stimulus. In our revised model (described below), we show how FB circuitry could read out h Δ C activity to produce upwind orientation during odor, reproducible walking in an arbitrary direction during sparse optogenetic activation, and unstable reorientation during uniform h Δ C activation. We believe these data and model provide a more complete picture of the role of h Δ C activity in olfactory navigation.

2) Model revisions.

All reviewers also noted issues with the model in our previous submission. Reviewer 1 was concerned that the model was presented as a definite circuit model rather than a more abstract conceptual model. Reviewer 2 asked for clarifications of the model logic, while Reviewer 3 expressed confusion about the role of the heading signal and suggested we include a simulated agent based on the model. In this submission, we have incorporated all of these suggestions to present a conceptual model, based on connectome motifs, that integrates h Δ C activity with heading information to drive a simulated agent. To explicitly connect our model with our experimental data, we modeled 3 different types of h Δ C activity: odor-gated wind direction-tuned activity based on our imaging data, sparse random optogenetic activation mimicking our sparse activation experiments, and uniform activation at various light levels mimicking uniform optogenetic activation. Our new model shows how motifs within the FB can read out these different patterns of h Δ C activity to produce three different behavioral phenotypes, consistent with our experimental results. We have also substantially revised the visual and text

presentation of our model to make its mechanisms clearer and connect it to previous modeling efforts. We believe this new model clarifies our hypotheses about how h Δ C activity can drive a goal heading, and suggests numerous follow-up experiments.

3) Analysis of imaging data

Reviewer 2 raised concerns about our statement that odor and wind pathways are segregated, drawing our attention to responses occurring at different phases of the stimulus such as wind onset or odor offset (end of the odor stimulus). To address this, we have added a classification analysis of neurons in the putative “odor” and “wind” pathways identified here and in our previous a manuscript (Currier et al. 2020). Using this analysis, we show that for the odor response, wind pathway neurons can be used to decode wind direction while odor pathway neurons cannot. In contrast, many odor pathway neurons can be used to decode odor from wind onset, while wind pathway neurons cannot. We also apply this analysis to other phases of the stimulus identifying one neuron group and stimulus phase in the “odor pathway” (wind onset in AD1b2/LH1396 neurons) that allows for significant decoding of wind direction. Looking more closely at these responses, we found that they were due to slightly larger responses to frontal wind versus wind from the sides. Based on these analysis we have updated our results to most accurately reflect our observations. On balance, we find that this classification analysis supports a separation of wind and odor encoding in separate anatomical pathways.

Reviewer 3 also raised a number of concerns about our analysis of calcium responses in h Δ C neurons and our claims about integration of odor and wind information in these cells. In our revised Fig 5 and S5, we include new analyses of h Δ C activity across stimulus phases and columns. We find that our results are most consistent with a bump of wind direction tuned activity that can be modulated by odor but also by other stimulus phases. We detail caveats with this interpretation, including the need for further experiments, in the Discussion. We also clarify that we do not believe h Δ C neurons are the only point of wind and odor integration in the fly brain, nor the only FB neurons involved in olfactory navigation. We hope that these revisions address the reviewer’s concerns, and acknowledge that future experiments, using more sophisticated experimental approaches, will be required to fully understand the nature of h Δ C representations and their behavioral role.

Below we address each of the reviewers’ concerns directly. (Reviewer comments in blue).

REVIEWER COMMENTS

Reviewer #1 (Remarks to the Author):

Summary

In this paper Matheson et al show that non-directional information about odors is combined with directional information about wind direction in the fan shaped body (FB) of the central complex (CX) in flies. This paper delivers first data on a central hypotheses generated by the fruit fly connectome, namely that contextual information is combined with spatial information in the CX. This is highly important work and it is carried out in an exemplary manner. A wide range of methods is used to systematically narrow down the possible integration sites of directional and non directional information from the antennal lobes, to the mushroom body, lateral horn, and input neurons to the FB, to finally converge on specific interneurons of the FB, the h Δ C cells. Using connectomics data from the recently published hemibrain connectome, the authors

generate an hypothetical circuit that is consistent with their data and that yields a network that can generate upwind orientation in flies in the presence of an attractive odor, while downwind orientation without an odor. Such context dependent switches in behavior are an important principle of how brains make decisions in general, and illustrate how sensory information is used differently depending on learned or innate context.

While I am not a fly geneticist and thus cannot comment on the details of fly lines and potential pitfalls, potentially missing controls etc, the behavioral, physiological, and imaging work is of very high quality, presented clearly and in a transparent way.

My only concern is that the final model of the paper is presented as an actual neural circuit, rather than a conceptual model that can be used as a starting point for further studies. The presented circuit, as illustrated in figure 6, is only partially possible when recapitulating it in the fly connectome data. Firmly established connections need to be more clearly distinguished from idealized and simplified connections, especially if they are crucial to the central conclusions of the paper.

That said, the paper is a beautiful and important paper, well written and enjoyable to read. It provides an important contribution to the field and will serve as starting point for many lines of investigations that aim at linking neural circuits to context dependent behavioral choices in animal brains.

We thank the reviewer for his enthusiastic assessment of the paper and have taken his suggestion to present the model as a conceptual framework rather than a definite circuit model.

Major comments

1) Stimulation of the hDeltaC neurons leads to turns. While such a response is expected when stimulating the non directional neurons representing valence (their information is downstream combined with directional wind information), I do not clearly see how activation of the entire population of hDeltaC neurons would lead to directed turns. This is because in these neurons wind information is already supposed to be combined with odor information and only those columns that receive PFN input as well as odor input will be naturally activated when initiating a turn into the appropriate direction (if I understood the model correctly). If all these cells are activated at the same time, no obvious source of asymmetry would be present and steering-neurons on both sides of the brain would be activated. Maybe the authors could clarify what the patterns of activation in hDeltaC neurons look like when driving these cells artificially. How is turning in response to global activation of these cells compatible with the model presented later in the paper?

This is a very reasonable question which we hope to have addressed with our new sparse activation data and updated model. In our new data, we show that asymmetric activation of h Δ C neurons can drive persistent walking in a reproducible direction, while uniform activation drives unstable orientation. Our new model provides a hypothesis for how these two phenotypes could arise. Asymmetric activity in h Δ C provides a “goal” signal that can be read out by the PFL system to produce stable orientation, while uniform activity might activate a mutually inhibitory population leading to neural oscillations and therefore unstable re-orientation. Both of these hypotheses should be testable in the future using imaging and targeted activation and silencing.

2) The interpretation of the data and the construction of a circuit to drive navigation appears to not be fully supported by the connectomics data. While elegant in its formulation and easy to understand, the links between the involved neurons, in particular between PFNa, hDeltaC and PFL3 neurons are not very strong (or do not exist) and hence the conceptual model presented is an idealized oversimplification. As the model is consistent with the presented data (but see point 1 above), I do believe it yields a valid hypothesis that can serve as a starting point for future studies, but the authors need to rework the presentation in a way that allows the readers to distinguish established neural connections from simplifications. As it stands, the reader is left with the impression that the diagram in Figure 6A depicts an actual and complete circuit, which is not backed up by sufficient connectomics data. In particular I found the following contradictions with the hemibrain connectome

(Hulse et al 2021, mostly based on figure 33 and figure 33 supplement 1 of that paper):

- PFNa neurons (most strongly responding to wind input) have no direct connection to PFL3 (or PFN2) neurons.
- They also have no strong indirect connection to PFL3 neurons, but form a separate pathway towards PFL1 neurons. While a weak connection to hDeltaC exists, and these neurons are connected to PFL3 via FC2B cells, these connections are weak and do not present an obvious major pathway through the FB. At this level of connectivity, many potential pathways through the FB exist, especially across three sequential synapses.
- PFNp and PFNm cells are a heterogeneous population of neurons with many different subtypes that connect to different targets. Grouping them with one another and with the very different PFNa cells is a bit misleading and it should at least be mentioned that this group of cells can lead to multiple different pathways. PFNm neurons also have no direct connections to PFL3 cells.
- The only direct pathway between PFN and PFL3 cells is realized by PFNp neurons, yet only one of these types is connected to hDeltaC neurons: PFNp_c. Hence the direct and indirect pathway from figure 6A can only exist for PFNp_c (if the same cell type should serve both pathways). Is anything known if these cells respond to wind stimuli?

In summary, the hypothesized direct and indirect pathways exist only if PFN neurons are treated as a group, rather than the diverse set of cell types they are. Only a single cell type (PFNp_c) can serve both pathways equally, but this is not the neuron with the strongest wind response (if I remember this correctly) and the pathway does not stand out anatomically. If, alternatively, different neurons feed into the direct and indirect pathways, a nicely balanced model (as presented) is not easy to achieve. Whether such a pathway is still feasible could be shown by computational models that take into account the actual synaptic weights of the connections (in future work).

I suggest to clarify and rework the model part of the paper, so that it becomes more clear that it is a hypothesis that illustrates an important and novel concept, rather than an actually existing circuit.

We would like to thank the review for his comments. We have substantially reworked our model so it no longer depends directly on specific connections between PFNs, h Δ C neurons, and PFL3. In the current version of the model, h Δ C neurons indirectly provide input to the PFL3 system, which integrates h Δ C activity, compass neurons activity, and the activity of a second mutually inhibitory local neuron population, to drive turning and forward velocity. We remain agnostic here about how wind responses in h Δ C are computed. We present two models— one in which the h Δ C wind representation is allocentric (which would require additional rear wind

vectors that have not yet been described) and one in which the h Δ C wind representation is confined to the frontal hemisphere, which could be computed based on the known responses of wind-sensitive PFNs (PFNa,p, and m). We also tried to clarify in this version (both in the Results and Discussion) that the model represents a conceptual framework rather than a specific circuit model, although we do believe it makes testable predictions about activity in various circuit elements. We hope that these revisions address the reviewer's concerns.

Minor comments

line 130: unclear what is meant by 'offset'. The term offset is used in the CX literature to describe an initial phase difference of the head direction code between individual flies, but that is clearly not what is meant here.

We have changed this to OFF to differentiate the temporal end of the stimulus from a spatial offset or phase difference.

line 380: Is the assumption that the PFL3 neurons are inhibitory based on any data? If not, then this should be stated more clearly.

This is a good question. Data does not exist on the neurotransmitter phenotype of PFL3 neurons, but numerous computational models of steering circuits in the FB (Hulse et al., 2021, Goulard et al. 2021, Stone et al., 2017, Sun et al., 2021) make the assumption that PFL3 neurons are inhibitory. If we learn that PFL3 neurons are excitatory, then we would have to update the model by introducing a 180 degree shift in the upwind goal representation, potentially through another horizontal local FB population or a switch in the neurotransmitter phenotype of intermediate neurons between h Δ C neurons and PFL3 neurons. In the current version of the manuscript we do not draw attention to this assumption but merely posit that activation of right PFL3 neurons leads to right turns.

line 476: Cx should be CX

We have fixed this.

(Stanley Heinze)

Reviewer #2 (Remarks to the Author):

Although it has been known that animals use both odor and wind direction cues to navigate the olfactory environment, the underlying neural circuit mechanisms are poorly understood. This manuscript utilizes the strength of various technologies available in flies including quantitative behavioral analysis, in vivo physiology, optogenetic manipulation, EM level connectome analysis, computational modeling, and identifies a neuron type, hDeltaC neuron, that integrates the information about odor and wind direction. Because the upstream neurons mainly convey either odor or wind direction information, the authors conclude that hDeltaC neuron is a site of integration. However, some data do not support their claims and there are several substantial issues that need to be addressed.

We thank the reviewer for pushing us to more precisely and carefully quantify our claims. We have added several new analyses of the imaging data to investigate these points and also

refined our claims in the Results and Discussion to more accurately reflect what we think we can conclude.

Major comments

1. One of the major claims is that wind-direction tuned activity of hDeltaC neuron is gated on by an odor. However, although hDeltaC neuron responds more weakly to wind onset than to an odor (Fig. 5G), it responds more strongly to wind offset than to an odor for several wind directions, at least for the example neuron shown in Fig. S5A, suggesting that it can transmit information about changes in wind speed and direction without gating by an odor. Quantification of data for all the recorded cells and re-evaluation of the claim are warranted.

This is a reasonable point. While we find that h Δ C neurons respond most strongly during odor (new Fig 5I, S5D), they can also respond at several other phases of the stimulus. We have provided a new analysis of wind direction tuning across stimulus phases. This analysis suggests that wind tuning at other phases is similar (usually within 45°) of the tuning during odor (Fig 5J). We conclude that h Δ C neurons show wind direction tuned activity that is most strongly activated by odor, but can also appear during other stimulus phases. We have qualified our conclusions in the revised Results and Discussion.

2. Similarly, LH1396 neurons (Fig. S2E) and FB5AB neurons (Fig. S4D) seem to show distinct wind tuning at wind offset and onset, respectively. Therefore, although these cells may not exhibit significant directional tuning at the transition of wind and wind+odor, they are tuned to wind direction at wind onset and offset. Based on to the way the authors previously characterized PFN neurons as wind-encoding (Currier et al., eLife, 2020), these neurons should also be described as wind-encoding besides odor-encoding.

We have re-examined this point in detail by developing a classifier analysis to ask whether different phases of the response in “odor pathway neurons” (LH, MB, and FB tangential neurons) encode significant information about wind direction. Based on this analysis, the only neuron group/phase in the “odor pathway” that allows for decoding of wind direction above chance is the AD1B2 (LH1396) group at wind onset. Further examination of these data revealed that this is due to slightly higher responses to frontal wind than wind from the sides. In contrast, neurons in the “wind pathway” (PFNa, difference between right and left LNa) allow for significant decoding of wind direction during all stimulus phases, but do not allow for decoding of odor onset versus wind onset.

Taken together, we think that the minimal representation of wind direction in “odor pathway” neurons, particularly during odor, justifies our assertion that FB local neurons are likely the first major site of integration for wind direction and odor cues, among the areas we surveyed here. We consider other possible sites of integration in the Discussion.

3. In Fig. 5H, the responses of hDeltaC neurons are re-aligned to the maximally responsive direction, but the original data without alignment is useful to examine if there is a relationship between the column and the maximally responsive direction. If this relationship is different across flies like the relationship between wind direction and peak response (Fig. 5E and line 307-308), what additional mechanisms should be required to explain the contribution of these cells to consistent upwind navigation of the flies?

We have added an analysis of wind tuning by column to Fig S5C. This analysis shows no consistent relationship between column identity and preferred wind direction. We interpret this

result to mean that the representation of wind direction in h Δ C neurons is likely in the coordinates of the EPG heading compass, which differs across animals. This would be consistent with recent anatomical work showing that PFNs multiplex sensory information from the noduli with heading information from the compass (Hulse et al., Lyu et al), so that the resulting information in FB local neurons is in compass coordinates (Lyu et al., Lu et al). In our revised model, we show how a wind bump in compass coordinates could be read out by the PFL system to generate upwind orientation, even though the wind bump can point in different directions in different flies. We further explore two possible representations of wind direction in h Δ C neurons— one which is fully allocentric, as has recently been described for traveling direction in h Δ B neurons (Lyu et al. Lu et al.), and one which is expressed relative to the compass but only for frontal wind directions, which could be computed from the known responses of wind-sensitive PFNs. We further note in the revised Discussion that additional experiments will be required to fully understand the nature of wind and odor representations in h Δ C neurons.

4. Fig. 5I is meant to show the causal effect of hDeltaC neurons on the upwind navigation. However, based on the physiology as well as the circuit model built in Fig. 6, this experiment is problematic because optogenetic stimulation activates all the hDeltaC neurons regardless of the fluctuating wind direction that the fly is experiencing at each moment during the light application period. Accordingly, the outcome is a turning behavior that is hard to explain. A proper approach is local activation of certain columns using focal P2X2 activation (e.g. Fujiwara et al., Nat Neurosci, 2016; Green et al., Nature, 2017) or optogenetic Chrimson activation (e.g. Kim et al., Science 2017), that require a fly on the ball. Even though the entire hDeltaC neurons are manipulated, an alternative is inhibition (e.g. GtACR) experiment in the freely behaving arena where the loss of upwind navigation would indicate the necessity of these neurons.

We agree that activation of all h Δ C neurons is problematic. To address this concern, we have added substantial new activation data in which we used SPARC2 (Isaacman-Beck et al.) to sparsely activate subsets of h Δ C neurons. As we show in our new Figure 6, this stimulation paradigm generates reproducible and persistent heading orientations in some flies, which we do not observe in flies where the SPARC2 construct is driven by an empty-Gal4 driver. We interpret this result to mean that asymmetric activation of h Δ C neurons can encode a goal heading that drives a persistent orientation.

We have also included new analysis of GtACR silencing data, which was only included in the Supplement in the previous version. We have re-analyzed these data, finding that silencing of h Δ C neurons impairs upwind orientation during the later part of the odor stimulus, an effect that we do not observe with FB5AB silencing. We interpret this to mean the h Δ C activity is necessary for persistent upwind fixation, though not for the initial upwind turn. We speculate on possible locations for neurons involved in this initial turn in the Discussion.

While we agree that fly-on-the-ball experiments will be critical to fully understand the nature of h Δ C representations, these experiments require a paradigm totally different from anything we have used in this paper thus far. Therefore we think these experiments fall outside the scope of the paper.

5. In the model shown in Fig. 6, I believe it would be easier to follow the logic if the implicit details or neurons are added to the circuit. For example, how can panel A generate

asymmetrical activation of right and left PFL3? Who are the integrators of left PFN and right PFN (Fig. 6 and line 364)?

We have substantially changed the framework and presentation of the model.

Minor comments

6. A tuning index (the difference between response to stimuli from 90 deg and -90 deg) does not account the responses to stimuli from the other three directions. There are better indices that quantify the direction selectivity.

This is a reasonable point and we have replaced this analysis with the more in-depth classification analysis.

7. Line 309. Fig. S3B should be Fig. S5A.

We have fixed this.

8. Line 73 Huovalia et al., 2020 and line 187 Schulze et al. 2015 are not listed in the References.

We have added these.

9. Line 1313. Jung et al., 2015 reported the ACV-responses of olfactory receptor neurons and not projection neurons.

We selected projection neurons that are directly downstream of the ORNs that responded to ACV in Jung et al. We have added a clarification of this point to the figure legend.

Reviewer #3 (Remarks to the Author):

This a very interesting and convincing account of how odour signals reach the steering centres of the insect brain (the Central Complex, CX) and are integrated with directional signals from wind to drive behaviour. It nicely combines behavioural, anatomical, optogenetic (silencing and activation) and modelling studies to present an unusually complete picture, and will be of substantial significance in the field and beyond.

Specifically they: demonstrate optogenetic activation of receptor neurons can substitute for odours in promoting upwind navigation behaviour, and these neurons are required; show a subset of mushroom body and lateral horn neurons produce movement relative to the wind, and are responsive to odour but not wind direction; identify through screening tangential neurons in the fan-shaped body, downstream from the MB, that promote upwind behaviour and respond to odour; used the connectome to identify FB local neurons that receive inputs from these tangential cells and previously identified columnar neurons that respond to wind direction; use calcium imaging to characterise the response of these neurons and optogenetic activation to show they are involved in steering; and finally modelling to provide a plausible interpretation of the circuit function.

We thank the Reviewer for this enthusiastic appraisal.

The following points of criticism are suggestions to improve the clarity of presentation:

Abstract final sentence (and elsewhere) "spatial and non-spatial information" - I feel throughout the manuscript, "directional and non-directional" might be a more accurate terminology. E.g. the odour events experienced by an animal moving in and out of an odour plume are spatial information.

We have rewritten the abstract and changed "spatial and non-spatial" to "directional and non-directional" throughout the text.

Introduction

line 50: it is not obvious how vision can provide information about the prevailing wind. Is the idea that optic flow caused by the wind disturbing motion can be interpreted as wind direction? This is far from straightforward, requiring disambiguation from self-motion etc. It might be simpler to eliminate 'or vision' here and focus on mechanosensation, introducing the potential use of vision more clearly in the discussion.

We have cut the reference to vision from the introduction.

line 53 "where these two types of information are integrated is not known for any species" seems too strong, e.g., see Sánchez- Alcañiz, J. A., & Benton, R. (2017). Multisensory neural integration of chemical and mechanical signals. *Bioessays*, 39(8), 1700060.

This is fair. We have changed this sentence to "Where and how these two types of information are integrated to support navigation towards an odor source is not clear."

line 56 "precise roles are not clear" - it is arguable that the 'roles' for these neuropils in some particular behaviours have been shown at least as 'precisely' as is shown for the behaviour examined here. This introduction should also mention how the lateral accessory lobe has been closely implicated in odour steering behaviour in moths.

We have removed this statement.

Results

It might be clearer to use 'vinegar' as the short-hand for 'apple cider vinegar' rather than ACV.

We have made this change.

Line 146 "activation in absence of wind...produced offset search" - is there information on the effect of vinegar in the absence of wind, or is this not possible to control precisely enough to reveal a similar effect? It is an interesting observation that could be better followed up (e.g. in the model) that upwind orientation and offset search might be distinct behaviours, but it is not clear why these data suggest that "these two behaviours are driven by distinct but overlapping populations of olfactory glomeruli" e.g. it could be an effect of the different temporal dynamics of neural responses to vinegar and optogenetic activation.

In a previous paper, we presented odor while stabilizing the antennae with UV glue (Alvarez-Salvado et al. 2018). This also eliminates upwind movement while leaving the offset search behavior entirely intact. Together we think these two results argue that the two behaviors (upwind and search) can be evoked independently. While it is possible that they are driven by

different temporal dynamics, we think the tight time-locking of offset search to the stimulus end means it is more likely driven by a different glomerular activation pattern. We explore in detail how stimulus dynamics drive search behavior in Alvarez-Salvado et al. 2018.

Line 163, This section might be better if the distinction of odour gradient behaviour, involving reduced curvature, and odour behaviour in flow is introduced at the start.

We have added a sentence to the first paragraph of the introduction describing the differences between wind-guided olfactory navigation and gradient navigation.

Line 212 on, it is a little unclear why "to test specifically whether alpha'3 MBONs show wind direction tuning", the experiments did not use the same presentation of wind from different directions as the previous tests, but instead looked at the differential response of ipsilateral and contralateral neurons to an odour presented from one side. The figure ref here is incorrect (2F->2G)

The alpha'3 MBON is included in the MB052B line, however based on the Mamiya study we wished to investigate this neuron in further detail. These experiments were performed earlier in the project using a different set-up. We thought they were worth including for completeness but not re-running with new stimuli, as the experiments took over a month and would probably take longer to redo. We have moved these data to the supplement in this version.

line 299 "might gate synaptic output" - it seems quite important to be clear about the envisioned axo-axonic mechanism here (it is touched on in the discussion). 'Gating' of output is rather different to the 'integration' described in fig 5. What might be understood from the inconsistent wind direction tuning? line 315, it seems unclear whether h-delta-cell activation produces specifically wind-directed turning or just turning?

We have tried to clarify our hypotheses about h Δ C representations and their role in behavior in this revision. Briefly, we think our results are consistent with the idea that h Δ C represents wind direction in compass coordinates, which would explain why the representation is different in each fly. h Δ C neurons appear to be activated most strongly by odor, suggesting that odor can modulate (gate ON) this wind direction-tuned activity, but that h Δ C can also show wind direction tuned activity more weakly at other phases of the stimulus. In our revised model, we present two hypotheses for the format of the h Δ C wind representation— one allocentric, and one frontal but in compass coordinates. In the model we assume this representation only turns on during odor, but presumably it could turn on at other times as well. Whenever this representation is ON it will tend to promote upwind orientation, due to the readout mechanism in PFL neurons.

Model: it is potentially confusing that the illustration in 6E assumes that 'heading bump' of the fly (from EPGs) is oriented in the same direction as the wind. It is consequently not obvious that the "+45 degrees and -45 degrees" are anatomical shifts of this bump (which could be in any direction relative to the wind) by one column due to PFN anatomy, which is then subsequently modulated in amplitude by the wind direction. In 6E the two 'bumps' seem to be at +/-45 degrees relative to the FBN and I initially assumed they were 'fixed' in this way and had to read the model details to understand otherwise. In fact I am still unsure - is the PFN bump (before amplitude modulation) determined by the wind direction? In the model description line 818, the 'heading' of the fly produces the bump, but can this be arbitrary with respect to the wind? Can the bump itself have an arbitrary (but consistent) offset from the heading (as observed in flies)? Does the model still work under these conditions?

We have substantially revised the model and its presentation which we hope will clarify answers to these questions. In the current version of the model, PFL neurons read out the difference between the h Δ C bump and the compass bump to promote turning until these two bumps align. So long as the heading bump and the wind bump are in the same coordinates, the model will always drive the fly upwind, regardless of how the map is oriented relative to the fly. We think these two signals are likely to be in the same coordinates, because the wind bump is likely built from PFN responses which inherit their heading tuning from EPGs. For example, to construct the “frontal” h Δ C wind representation used in the previous version of this paper and used in Fig S7A-D of the revised paper, the PFN neurons have bumps of activity that shift with heading and scale with egocentric wind direction. Since the ‘wind’ bumps in h Δ C neurons and the ‘heading’ bumps in PFL3 and PFL2 neurons are both inherited from the same EPG compass, the relative overlap between wind and heading bumps will be maintained regardless of the compass tuning or the signals used to modulate EPG activity (e.g. the system would still work if wind signals are used to update the heading bump in EPG neurons). However, further experiments will be needed to test this directly.

Figure 6F also seems a bit misleading, showing what looks like an all-or-nothing amplitude modulation rather than clearly illustrating how the preferred tuning of the TN cells is assumed to modulate the amplitude.

We have substantially changed the model and figure. In the current version we assume that odor gates the h Δ C wind representation ON in an all-or-nothing manner. However, graded control of the wind bump amplitude is also possible. As we do not yet know a great deal about the tuning of FB5AB or the relationship between FB5AB amplitude and h Δ C amplitude we have kept this relationship simple in this model. However, this would be a fruitful avenue for future research.

The general story of the paper led me to expect that the model would illustrate the smooth integration of the spatial and non-spatial information, e.g. with a simulated agent exposed to wind and changing its behaviour in the presence of odour. But the model results shown are more static, just the independent steering output of the two putative pathways, with the assumption (not modelled) that the presence of odour switches the system into use of the indirect pathway (and switches off the direct pathway?). The model itself does not include any term for odour, or how the h-delta-C neurons integrate (or are gated) by it, or how (line 336) “competing pathways” “alter the balance of activity between” them based on odour.

We thank the Reviewer for this suggestion. In the current version of the model we have used it to drive a simulated agent and explicitly model the dynamics when odor (or optogenetic activation) are transiently turned on. We find that activity in h Δ C neurons promotes stable orientation for as long as the neurons are active. This could be driven by odor, by optogenetic activation, or due to spontaneous activation of h Δ C neurons.

It would be appropriate here or in the discussion to reference this paper:

Goulard R, Buehlmann C, Niven JE, Graham P, Webb B (2021) A unified mechanism for innate and learned visual landmark guidance in the insect central complex. PLOS Computational Biology 17(9): e1009383. <https://doi.org/10.1371/journal.pcbi.1009383>

- which provides a similar, but more subtle model for how a non-directional 'value' signal from the MB could be integrated with directional steering in the CX through PFN/PFL interaction.

We have added citations to this paper in several places.

Discussion

A good point is made about the separation of identity and directional information but seems easily tested whether learning of value in one behavioural task, e.g. walking or proboscis extension, generalises to another, e.g. flight choice decisions. Is there not any citable study of this?

We have added a citation to Chaffiol et al. 2005 in bees.

Around line 530, a plausible possibility is that, compared to direct steering pathways, the CX allows the animal to maintain steering in a particular direction even when the cue or stimulus for steering is intermittent. The Goulard et al. (2021) model mentioned above demonstrates both this and the ability to adopt a course at a particular angle to the directional cue (line 535).

This is an excellent point and we now discuss this in greater depth in lines 910-935.

REVIEWERS' COMMENTS

Reviewer #1 (Remarks to the Author):

The resubmitted paper by Matheson et al. is a greatly improved version of the original manuscript and all my concerns were addressed fully. The authors added not only a set of valuable experiments that validate the role of hdeltaC neurons in determining directional behavior in the presence of wind and odor, they also added a full-scale computational model that recapitulates all observations and makes exciting predictions about where directional memories could be stored in the CX. The model assumes an additional circuit motive (mutual inhibition) that was not directly observed in the connectome, but which is plausible.

The authors are also more careful in making sure that the model is perceived as a model and not as fact. They highlight simplifications from the biological circuits and are transparent about limitations.

Importantly, the new biological data allows more far reaching conclusions than the original manuscript. I find those conclusions very exciting and, while similar thoughts were implied in some recent papers on the CX, the explicit, and data driven statement below, summarizes nicely where I personally see huge capacity for future research:

"One advantage of our model is that different local neurons can be rapidly switched on and off through tangential input, allowing the fly to rapidly update its goals depending on behavioral demands. A further advantage is the large number of local neurons and tangential inputs, which as noted above provide a substrate for learning, storing, and releasing multiple goal memories."

With the wealth of data and important new concepts, this paper is a highly important contribution to the field!

one tiny edit: line 207: Cx should be CX

Reviewer #2 (Remarks to the Author):

The authors have addressed all the issues previously raised by this reviewer. I would like to congratulate the authors for their contribution to the field through this beautiful manuscript.

Regarding the revised manuscript, I was a bit confused by the rather sudden shift in description of hdeltaC coding in the final modelling section. Until then, hdeltaC was described to encode odor-gated wind direction and it is labeled in such a way in Figure 7C as well. However, from line 362, they start to describe that hdeltaC encodes "goal" , which never explicitly appears in the study.

line 362

PFL neurons compare the "goal" representation arriving from hΔC neurons (a bump of activity flipped by 180° due to hΔC neurons projecting to columns halfway across the FB) with a shifted representation of the fly's current heading arriving from compass neurons (Stone et al. 2017, Hulse et al. 2021)."

Although the final output of the model may be regarded as "goal-directed/oriented behavior", there is no problem in explaining the model by sticking to the same description that hdeltaC encodes wind direction gated by an odor. The connection between the activity of hdeltaC and goal-directed behavior can be speculated in Discussion, but I feel that it is easier for the readers to follow the manuscript without this shift in description.

Reviewer #3 (Remarks to the Author):

This revised paper shows a commendable effort to respond to the previous reviews, including new data and an enhanced model to support the key arguments. As before, I consider the work both highly convincing and extremely interesting. The following are minor comments/questions that the authors might consider in finalising the manuscript, rather than required revisions.

Line 80 "FB circuitry is optimized for encoding vectors..." could additionally reference Le Moël, Florent, et al. "The central complex as a potential substrate for vector based navigation." *Frontiers in psychology* 10 (2019): 690

Line 266 on, I am not sure my previous comment on this issue has been fully understood as the response seems a bit beside the point. The anatomy suggests that the influence of odor from FB5AB on h-delta-c cells is axo-axonic. This is (to my mind) an important difference from a dendritic input, particularly as it suggests 'gating' of the output rather than 'integration' of two inputs, yet the authors seem inclined to

use 'gating' and 'integration' interchangeably (e.g. in caption to fig 8, "columnar and tangential inputs...are integrated by local neurons..."). Specifically, it suggests, h-delta-c cells are NOT 'activated most strongly by odor' (as written in response to reviewers) but rather, are activated by wind, and then their output (towards PFLs specifically?) is modulated by odor. It is then relevant to highlight that the recordings from h-delta-c that show odor modulation in activity are from calcium recordings at the output tufts (line 273). This distinction perhaps does not matter for the simplified model (see below) but seems worth making more explicit mention of this, as, for example, the existence and potential function of axo-axonic connections is to date almost completely ignored in artificial neural networks and computational neuroscience.

Fig 5C shows the pattern of synapse numbers between different wind PFN and h-delta-c per column (actually, why are there 12 rather than 16 columns shown here?). But I could not locate in the text any discussion of how this pattern might be related to the h-delta-c wind tuning. Is it relevant? Is this meant to underlie the alternative h-delta-c model in S7A?

Line 348 on, it seems somewhat inconsistent to argue that the 'bump' in the h-delta-c neurons is reflecting fly-specific coordinates of the EPG compass but can be represented as reflecting the allocentric wind direction, meaning that it is independent of the compass. I don't think there is any evidence in this paper per se to support the allocentric model? If I understand correctly, the implementation of this version of model does not actually include the arbitrary offset of the compass as seen in flies, or equivalently, requires any arbitrary offset is the same for both the compass and the allocentric wind direction? It is good that an alternative model somewhat more grounded in the possible connectivity that gives rise to the h-delta-c bump is included, but it seems this alternative model does not work as well, and no results are shown for the alternative model under the condition of sparse activation of h-delta-c neurons (fig S7). It is also slightly disappointing that the 'model' of modulation by odor inputs remains a simple all-or-nothing effect (multiplication by a 'gain' term that can only be 0 or 1). However it seems reasonable to leave this to future work.

Line 384 "network of firing rate-based neurons" is confusing terminology, I assume they mean [firing rate]-based, but it reads as firing [rate-based]. The standard terminology would just be 'rate-based neurons'.

Responses to Reviewers

Please see below our responses to the remaining reviewer comments. Our responses in blue.

REVIEWERS' COMMENTS

Reviewer #1 (Remarks to the Author):

The resubmitted paper by Matheson et al. is a greatly improved version of the original manuscript and all my concerns were addressed fully. The authors added not only a set of valuable experiments that validate the role of hdeltaC neurons in determining directional behavior in the presence of wind and odor, they also added a full-scale computational model that recapitulates all observations and makes exciting predictions about where directional memories could be stored in the CX. The model assumes an additional circuit motive (mutual inhibition) that was not directly observed in the connectome, but which is plausible.

The authors are also more careful in making sure that the model is perceived as a model and not as fact. They highlight simplifications from the biological circuits and are transparent about limitations.

Importantly, the new biological data allows more far reaching conclusions than the original manuscript. I find those conclusions very exciting and, while similar thoughts were implied in some recent papers on the CX, the explicit, and data driven statement below, summarizes nicely where I personally see huge capacity for future research:

"One advantage of our model is that different local neurons can be rapidly switched on and off through tangential input, allowing the fly to rapidly update its goals depending on behavioral demands. A further advantage is the large number of local neurons and tangential inputs, which as noted above provide a substrate for learning, storing, and releasing multiple goal memories."

With the wealth of data and important new concepts, this paper is a highly important contribution to the field!

We thank the reviewer for his very helpful feedback and kind words about our paper!

one tiny edit: line 207: Cx should be CX

fixed

Reviewer #2 (Remarks to the Author):

The authors have addressed all the issues previously raised by this reviewer. I would like to congratulate the authors for their contribution to the field through this beautiful manuscript.

Thank you for the kind assessment of our manuscript.

Regarding the revised manuscript, I was a bit confused by the rather sudden shift in description of hdeltaC coding in the final modelling section. Until then, hdeltaC was described to encode odor-gated wind direction and it is labeled in such a way in Figure 7C as well. However, from line 362, they start to describe that hdeltaC encodes "goal", which never explicitly appears in the study.

line 362

PFL neurons compare the "goal" representation arriving from h Δ C neurons (a bump of activity flipped by 180° due to h Δ C neurons projecting to columns halfway across the FB) with a shifted representation of the fly's current heading arriving from compass neurons (Stone et al. 2017, Hulse et al. 2021)."

Although the final output of the model may be regarded as "goal-directed/oriented behavior", there is no problem in explaining the model by sticking to the same description that hdeltaC encodes wind direction gated by an odor. The connection between the activity of hdeltaC and goal-directed behavior can be speculated in Discussion, but I feel that it is easier for the readers to follow the manuscript without this shift in description.

We have removed the word "goal" from this paragraph.

Reviewer #3 (Remarks to the Author):

This revised paper shows a commendable effort to respond to the previous reviews, including new data and an enhanced model to support the key arguments. As before, I consider the work both highly convincing and extremely interesting. The following are minor comments/questions that the authors might consider in finalising the manuscript, rather than required revisions.

Line 80 "FB circuitry is optimized for encoding vectors..." could additionally reference Le Moël, Florent, et al. "The central complex as a potential substrate for vector based navigation." *Frontiers in psychology* 10 (2019): 690

We have added this citation.

Line 266 on, I am not sure my previous comment on this issue has been fully understood as the response seems a bit beside the point. The anatomy suggests that the influence of odor from FB5AB on h-delta-c cells is axo-axonic. This is (to my mind)

an important difference from a dendritic input, particularly as it suggests 'gating' of the output rather than 'integration' of two inputs, yet the authors seem inclined to use 'gating' and 'integration' interchangeably (e.g. in caption to fig 8, "columnar and tangential inputs...are integrated by local neurons..."). Specifically, it suggests, h-delta-c cells are NOT 'activated most strongly by odor' (as written in response to reviewers) but rather, are activated by wind, and then their output (towards PFLs specifically?) is modulated by odor. It is then relevant to highlight that the recordings from h-delta-c that show odor modulation in activity are from calcium recordings at the output tufts (line 273). This distinction perhaps does not matter for the simplified model (see below) but seems worth making more explicit mention of this, as, for example, the existence and potential function of axo-axonic connections is to date almost completely ignored in artificial neural networks and computational neuroscience.

We agree that the finding of FB5AB modulating h Δ C outputs is intriguing and likely relevant to how the system selects navigational outputs. Given the space constraints of the paper we do not think we can fully explore this idea here. However, we have made a few text changes to emphasize this idea: We have added the phrase "through axo-axonic connections" at line 266. We have also changes the last sentence of the Fig. 8 caption to read: "and where tangential input can gate the expression of directional information in local neurons outputs to specify navigational goals."

Fig 5C shows the pattern of synapse numbers between different wind PFN and h-delta-c per column (actually, why are there 12 rather than 16 columns shown here?). But I could not locate in the text any discussion of how this pattern might be related to the h-delta-c wind tuning. Is it relevant? Is this meant to underlie the alternative h-delta-c model in S7A?

Our point here was simply that each h Δ C cell receives input from both left and right-tuned PFNs, giving it the capacity to respond to any wind direction. We have added the phrase "from both left and right-preferring wind-sensitive PFNs" to the results where we introduce this figure panel. h Δ C is shown to have 12 columns in Hulse et al. 2020.

Line 348 on, it seems somewhat inconsistent to argue that the 'bump' in the h-delta-c neurons is reflecting fly-specific coordinates of the EPG compass but can be represented as reflecting the allocentric wind direction, meaning that it is independent of the compass. I don't think there is any evidence in this paper per se to support the allocentric model? If I understand correctly, the implementation of this version of model does not actually include the arbitrary offset of the compass as seen in flies, or equivalently, requires any arbitrary offset is the same for both the compass and the allocentric wind direction? It is good that an alternative model somewhat more grounded in the possible connectivity that gives rise to the h-delta-c bump is included, but it seems this alternative model does not work as well, and no results are shown for the alternative model under the condition of sparse activation of h-delta-c neurons (fig S7). It is also slightly disappointing that

the 'model' of modulation by odor inputs remains a simple all-or-nothing effect (multiplication by a 'gain' term that can only be 0 or 1). However it seems reasonable to leave this to future work.

Our goal here was to explore how different wind representations in h Δ C might influence navigation. Given recent studies showing allocentric representations in h Δ B neurons, we thought it was reasonable to ask how an allocentric wind representation might influence navigation. Such a representation would require "backward" vectors for wind direction which have not yet been described, though they could be present in other PFN types. We agree that there are many more variations of the model which would be interesting to explore but we agree with the reviewer that this would be best left to future work.

Line 384 "network of firing rate-based neurons" is confusing terminology, I assume they mean [firing rate]-based, but it reads as firing [rate-based]. The standard terminology would just be 'rate-based neurons'.

We have changed this to "rate based neurons" as suggested.